# VolleyBots: A Testbed for Multi-Drone Volleyball Game Combining Motion Control and Strategic Play

**Zelai Xu**[1][*], **Ruize Zhang**[2][*], **Chao Yu**[2][†], **Huining Yuan**[2], **Xiangmin Yi**[2], **Shilong Ji**[2], **Chuqi Wang**[1], **Wenhao Tang**[2], **Feng Gao**[3], **Wenbo Ding**[2], **Xinlei Chen**[2], **Yu Wang**[1][†]

[1]EE, Tsinghua University   [2]SIGS, Tsinghua University   [3]IIIS, Tsinghua University

## Abstract

Robot sports, characterized by well-defined objectives, explicit rules, and dynamic interactions, present ideal scenarios for demonstrating embodied intelligence. In this paper, we present *VolleyBots*, a novel robot sports testbed where multiple drones cooperate and compete in the sport of volleyball under physical dynamics. VolleyBots integrates three features within a unified platform: *competitive and cooperative gameplay*, *turn-based interaction structure*, and *agile 3D maneuvering*. These intertwined features yield a complex problem combining motion control and strategic play, with no available expert demonstrations. We provide a comprehensive suite of tasks ranging from single-drone drills to multi-drone cooperative and competitive tasks, accompanied by baseline evaluations of representative reinforcement learning (RL), multi-agent reinforcement learning (MARL) and game-theoretic algorithms. Simulation results show that on-policy RL methods outperform off-policy methods in single-agent tasks, but both approaches struggle in complex tasks that combine motion control and strategic play. We additionally design a hierarchical policy which achieves 69.5% win rate against the strongest baseline in the *3 vs 3* task, demonstrating its potential for tackling the complex interplay between low-level control and high-level strategy. To highlight VolleyBots' sim-to-real potential, we further demonstrate the zero-shot deployment of a policy trained entirely in simulation on real-world drones.

⊙ **Benchmark & Code:** https://github.com/thu-uav/VolleyBots
⊛ **Project Website:** https://sites.google.com/view/thu-volleybots

## 1 Introduction

Robot sports, characterized by their well-defined objectives, explicit rules, and dynamic interactions, provide a compelling domain for evaluating and advancing embodied intelligence. These scenarios require agents to effectively integrate real-time perception, decision-making, and control in order to accomplish specific goals within physically constrained environments. Several existing efforts have explored such environments: robot football [1, 2, 3, 4] emphasizes both intra-team cooperation and inter-team competition; robot-arm table tennis [5, 6] features the turn-based nature of ball exchange; and multi-drone pursuit-evasion [7] demands agile maneuvering in a 3D space.

In this work, we introduce a novel robot sports testbed named ***VolleyBots***, where multiple drones engage in the popular sport of volleyball. *VolleyBots integrates all these three key features into a unified platform: mixed competitive and cooperative game dynamics, a turn-based interaction structure, and agile 3D maneuvering. Mixed competitive and cooperative game dynamics* necessitates

---

[*] Equal contribution. `{zelai.eecs,jimmyzhangruize}@gmail.com`

[†] Corresponding authors. `{yuchao,yu-wang}@mail.tsinghua.edu.cn`

39th Conference on Neural Information Processing Systems (NeurIPS 2025) Track on Datasets and Benchmarks.

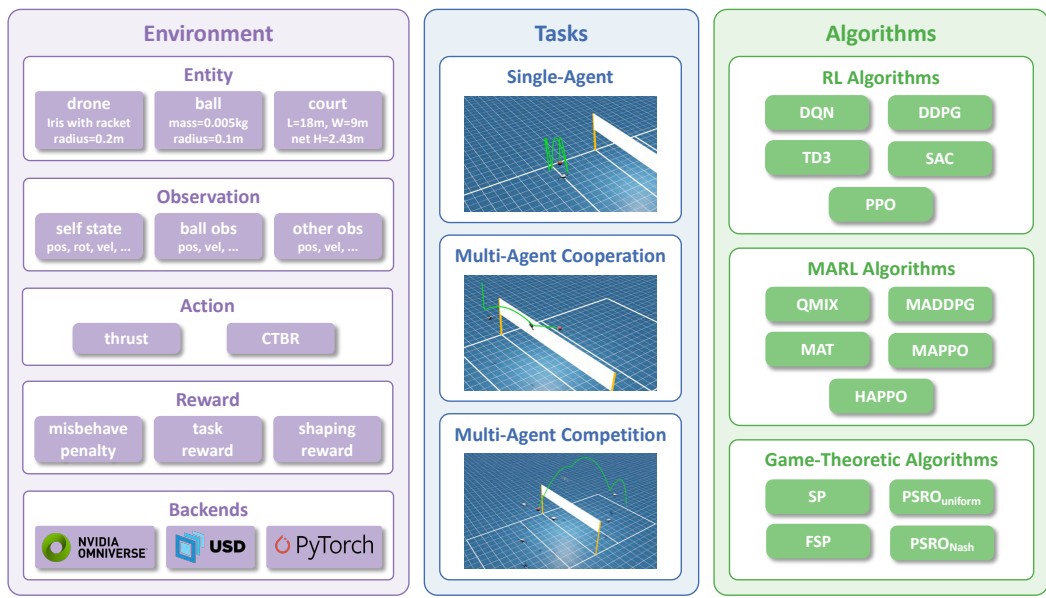

Figure 1: Overview of the VolleyBots Testbed. VolleyBots comprises three components: (1) Environment, supported by Isaac Sim, which defines entities, observations, actions, and reward functions; (2) Tasks, including 3 single-agent tasks, 3 multi-agent cooperative tasks, and 3 multi-agent competitive tasks; and (3) Algorithms, encompassing RL, MARL, and game-theoretic algorithms.

that each drone achieves tight coordination with teammates, enabling intra-team passing sequences. Simultaneously, each team must proactively anticipate and effectively exploit the offensive and defensive strategies of opposing agents. *Turn-based interaction* in VolleyBots operates on two levels: inter-team role switching between offense and defense, and intra-team coordination for ball-passing sequences. This dual-level structure demands precise timing, accurate state prediction, and effective management of long-horizon temporal dependencies. *Agile 3D maneuvering* demands that each drone performs rapid accelerations, sharp turns, and fine-grained positioning, all while operating under the underactuated quadrotor dynamics. This challenge is intensified by frequent contacts with the ball, which disrupt the drone's orientation and require post-contact recovery to maintain control. These intertwined features not only create a challenging problem that combines motion control and strategic play, but also lead to the absence of expert demonstrations.

The overview of the VolleyBots testbed is shown in Fig. 1. Built on Nvidia Isaac Sim [8], VolleyBots supports efficient GPU-based data collection. Inspired by how humans progressively learn the structure of volleyball, we design a curriculum of tasks ranging from single-drone drills to multi-drone cooperative plays and competitive matchups. We have also implemented reinforcement learning (RL), multi-agent reinforcement learning (MARL) and game-theoretic baselines, and provided benchmark results. In single-agent tasks, simulation results show that with a single set of hyperparameters, on-policy RL methods maintains consistently strong performance across multiple tasks, demonstrating superior robustness compared to off-policy methods. However, both approaches struggle in more complex tasks that require low-level motion control and high-level strategic play. To demonstrate real-world deployment ability, we show a policy trained to bump volleyball can be deployed on an open-source quadrotor equipped with a racket in a zero-shot manner. We envision VolleyBots as a valuable platform for advancing the study of embodied intelligence in physically grounded, multi-agent robotic environments.

Our main contributions are summarized as follows:

1. We introduce VolleyBots, a novel robot sports environment centered on drone volleyball, featuring mixed competitive and cooperative game dynamics, turn-based interactions, and agile 3D maneuvering while demanding both motion control and strategic play.

2. We release a curriculum of tasks, ranging from single-drone drills to multi-drone cooperative plays and competitive matchups, and baseline evaluations of representative MARL and game-theoretic algorithms, facilitating reproducible research and comparative assessments.

3. We design a hierarchical policy that achieves a 69.5% win rate against the strongest baseline in the *3 vs 3* task, offering a promising solution for tackling the complex interplay between low-level control and high-level strategy.

## 2  Related work

### 2.1  Robot sports

The integration of sensing, actuation, and autonomy has enabled a wide range of robotic platforms, spanning robotic arms, quadrupeds, humanoids, and aerial drones, to undertake increasingly complex tasks. Robot sports provide a compelling testbed for evaluating their capabilities within well-defined rule sets. A classic example is robot soccer: since the initiative of RoboCup [9], research into autonomous football has driven advances in multi-agent coordination, strategic planning, and hardware integration. Early approaches [10, 11] to robot sports relied primarily on classical control and planning techniques. With the growth of data-driven methods, imitation learning algorithms [12] enabled robots to learn complex motion policies directly from expert demonstrations. More recently, RL methods have also achieved remarkable performance. With RL, Researchers have explored a wide range of robot platforms for sports tasks. Robot arms on mobile bases have learned table tennis [5, 6] and badminton [13]. Quadrupeds have commanded basic soccer drills [2] and played in multi-agent football matches [4]. Humanoid robots have demonstrated competitive 1 vs 1 [3] and 2 vs 2 [1] football skills and participated in simulated Olympic-style events SMPLOlympics [14]. Drones have achieved human-surpassing racing performance [15] and tackled multi-UAV pursuit-evasion tasks with rule-based pursuit policies [7]. Despite these advances, there remains a need for environments that combine high-mobility platforms (e.g., drones) with mixed cooperative-competitive dynamics and require both high-level decision-making and low-level continuous control. To fill this gap, we introduce VolleyBots, a turn-based, drone-focused sports environment that seamlessly integrates strategic planning with agile control. Built on a realistic physics simulator, VolleyBots offers a unique testbed for advancing research in agile, decision-driven robot control. A detailed comparison between VolleyBots and representative learning-based robot sports platforms is provided in Table 1.

Table 1: Comparison of VolleyBots and existing representative learning-based robot sports works.

| | Multi-Agent Task | | | Game Type | Entity | Hierarchical Policy | Open Source | Baseline Provided |
| --- | --- | --- | --- | --- | --- | --- | --- | --- |
| | coop. | comp. | mixed | | | | | |
| Robot Table Tennis [5] | ✗ | ✓ | ✗ | turn-based | robotic arm | ✓ | ✗ | ✗ |
| Badminton Robot [13] | ✗ | ✗ | ✗ | turn-based | robotic arm | ✗ | ✗ | ✗ |
| Quadruped Soccer [2] | ✗ | ✗ | ✗ | simultaneous | quadruped | ✓ | ✗ | ✗ |
| MQE [4] | ✓ | ✓ | ✓ | simultaneous | quadruped | ✓ | ✓ | ✓ |
| Humanoid Football [1] | ✗ | ✓ | ✓ | simultaneous | humanoid | ✓ | ✓ | ✗ |
| SMPLOlympics [14] | ✗ | ✓ | ✓ | simu. & turn-based | humanoid | ✗ | ✓ | ✓ |
| Pursuit-Evasion [7] | ✓ | ✗ | ✗ | simultaneous | drone | ✗ | ✓ | ✓ |
| Drone-Racing [15] | ✗ | ✗ | ✗ | simultaneous | drone | ✗ | ✗ | ✗ |
| VolleyBots (Ours) | ✓ | ✓ | ✓ | turn-based | drone | ✓ | ✓ | ✓ |

### 2.2  Learning-based methods for drone control task

Executing precise and agile flight maneuvers is essential for drones, which has driven the development of diverse control strategies [16, 17, 18]. While traditional model-based controllers excel in predictable settings, learning-based approaches adapt more effectively to dynamic, unstructured environments. One popular approach is imitation learning [19, 20], which trains policies from expert demonstrations. However, collecting high-quality expert data, especially for aggressive or novel maneuvers, can be costly or infeasible. In such a case, RL offers a flexible alternative by discovering control policies through trial-and-error interaction. Drone racing is a notable single-drone control task where RL has achieved human-level performance [21], showcasing near-time-optimal decision-making capabilities. Beyond racing, researchers also leveraged RL for executing aggressive flight maneuvers [22] and achieving hovering stabilization under highly challenging conditions [18]. As

for multi-drone tasks, RL has been applied to cooperative tasks such as formation maintenance [23], as well as more complex scenarios like multi-drone pursuit-evasion tasks [7], further showcasing its potential to jointly optimize task-level planning and control. In this paper, we present VolleyBots, a testbed designed to study the novel drone control task of drone volleyball. This task introduces unique challenges, requiring drones to learn both cooperative and competitive strategies at the task level while maintaining agile and precise control. Additionally, VolleyBots provides a comprehensive platform with (MA)RL and game-theoretic algorithm baselines, facilitating the development and evaluation of advanced drone control strategies.

## 3 VolleyBots environment

In this section, we introduce the environment design of the VolleyBots testbed. The environment is built upon the high-throughput and GPU-parallelized OmniDrones [24] simulator, which relies on Isaac Sim [8] to facilitate rapid data collection. We further configure OmniDrones to simulate realistic flight dynamics and interaction between the drones and the ball, then implement standard volleyball rules and gameplay mechanics to create a challenging domain for drone control tasks. We will describe the simulation entity, observation space, action space, and reward functions in the following subsections.

### 3.1 Simulation entity

Our environment simulates real-world physics dynamics and interactions of three key components including the drones, the ball, and the court. We provide a flexible configuration of each entity's model and parameters to enable a wide range of task designs. For the default configuration, we adopt the *Iris* quadrotor model [25] as the primary drone platform, augmented with a virtual "racket" of radius $0.2\,\mathrm{m}$ and coefficient of restitution $0.8$ for ball striking. The ball is modeled as a sphere with a radius of $0.1\,\mathrm{m}$, a mass of $5\,\mathrm{g}$, and a coefficient of restitution of $0.8$, enabling realistic bounces and interactions with both drones and the environment. The court follows standard volleyball dimensions of $9\,\mathrm{m} \times 18\,\mathrm{m}$ with a net height of $2.43\,\mathrm{m}$.

### 3.2 Observation space

To align with the feature of partial observability in real-world volleyball games, we adopt a state-based observation space where each drone can fully observe its own physical state and partially observe the ball's state and other drones' states. More specifically, each drone has full observability of its position, rotation, velocity, angular velocity, and other physical states. For ball observation, each drone can only partially observe the ball's position and velocity. In multi-agent tasks, each drone can also partially observe other drones' positions and velocities. Minor variations in the observation space may be required for different tasks, such as the ID of each drone in multi-agent tasks. Detailed observation configurations for each task are provided in the Appendix D.

### 3.3 Action space

We provide two types of continuous action spaces that differ in their level of control, with Collective Thrust and Body Rates (CTBR) offering a higher-level abstraction and Per-Rotor Thrust (PRT) offering a more fine-grained manipulation of individual rotors.

**CTBR.** A typical mode of drone control is to specify a single collective thrust command along with body rates for roll, pitch, and yaw. This higher-level abstraction hides many hardware-specific parameters of the drone, often leading to more stable training. It also simplifies sim-to-real transfer by reducing the reliance on precise modeling of individual rotor dynamics.

**PRT.** Alternatively, the drone can directly control each rotor's thrust individually. This fine-grained control allows the policy to fully exploit the drone's agility and maneuverability. However, it typically requires a more accurate hardware model, making system identification more complex, and can increase the difficulty of sim-to-real deployment.

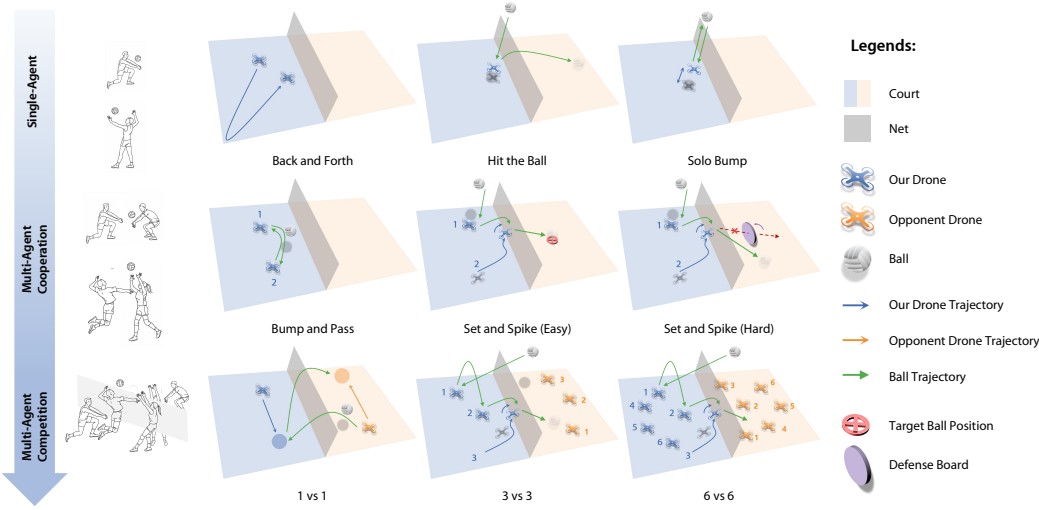

Figure 2: Proposed tasks in the VolleyBots testbed, inspired by the process of human learning in volleyball. Single-agent tasks evaluate low-level control, while multi-agent cooperative and competitive tasks integrate high-level decision-making with low-level control.

## 3.4 Reward functions

The reward function for each task consists of three parts, including the misbehave penalty for general motion control, the task reward for task completion, and the shaping reward to accelerate training.

**Misbehave penalty.** This term is consistent across all tasks and penalizes undesirable behaviors related to general drone motion control, such as crashes, collisions, and invalid hits. By imposing penalties for misbehavior, the drones are guided to maintain physically plausible trajectories and avoid actions that could lead to control failure.

**Task reward.** Each task features a primary objective-based reward that encourages the successful completion of the task. For example, in solo bump tasks, the drone will get a reward of 1 for each successful hit of the ball. Since the task rewards are typically sparse, agents must rely on effective exploration to learn policies that complete the task.

**Shaping reward.** Due to the sparse nature of many task rewards, relying solely on the misbehave penalty and the task reward can make it difficult for agents to successfully complete the tasks. To address this challenge, we introduce additional shaping rewards to help steer the learning process. For example, the drone's movement toward the ball is rewarded when a hit is required. By providing additional guidance, the shaping rewards significantly accelerate learning in complex tasks.

## 4 VolleyBots tasks

Inspired by the way humans progressively learn to play volleyball, we introduce a series of tasks that systematically assess both low-level motion control and high-level strategic play, as shown in Fig. 2. These tasks are organized into three categories: single-agent, multi-agent cooperative, and multi-agent competitive. Each category aligns with standard volleyball drills or match settings commonly adopted in human training, ranging from basic ball control, through cooperative play, to competitive full games. Evaluation metrics vary across tasks to assess performance in motion control, cooperative teamwork, and strategic competition. The detailed configuration and reward design of each task can be found in Appendix D.

Table 2: Benchmark result of single-agent tasks with different action spaces including Collective Thrust and Body Rates (CTBR) and Per-Rotor Thrust (PRT). *Back and Forth* is evaluated by the number of target points reached, *Hit the Ball* is evaluated by the hitting distance, and *Solo Bump* is evaluated by the number of bumps achieving a certain height.

| | *Back and Forth* | | *Hit the Ball* | | *Solo Bump* | |
| | CTBR | PRT | CTBR | PRT | CTBR | PRT |
| --- | --- | --- | --- | --- | --- | --- |
| DQN | $0.00 \pm 0.00$ | $0.00 \pm 0.00$ | $0.39 \pm 0.02$ | $1.88 \pm 0.34$ | $0.00 \pm 0.00$ | $0.00 \pm 0.00$ |
| DDPG | $1.14 \pm 0.34$ | $0.83 \pm 0.23$ | $2.87 \pm 0.55$ | $3.98 \pm 1.08$ | $0.44 \pm 0.34$ | $0.67 \pm 0.32$ |
| TD3 | $1.12 \pm 0.68$ | $0.99 \pm 0.01$ | $3.00 \pm 0.52$ | $3.91 \pm 0.35$ | $3.68 \pm 1.43$ | $5.29 \pm 1.28$ |
| SAC | $0.90 \pm 0.12$ | $0.83 \pm 0.25$ | $3.76 \pm 1.46$ | $3.87 \pm 2.34$ | $0.54 \pm 0.27$ | $1.36 \pm 0.60$ |
| PPO | $\mathbf{9.25 \pm 0.31}$ | $\mathbf{10.04 \pm 0.20}$ | $\mathbf{10.48 \pm 0.08}$ | $\mathbf{11.40 \pm 0.06}$ | $\mathbf{8.58 \pm 0.79}$ | $\mathbf{10.83 \pm 1.24}$ |

## 4.1 Single-agent tasks

Single-agent tasks are designed to follow typical solo training drills used in human volleyball practice, including *Back and Forth*, *Hit the Ball*, and *Solo Bump*. These tasks evaluate the drone's agile 3D maneuvering capabilities, such as flight stability, motion control, and ball-handling proficiency.

***Back and Forth.*** The drone sprints between two designated points to complete as many round trips as possible within the time limit. This task is analogous to the back-and-forth sprints in volleyball practice. The rapid acceleration, deceleration, and precise altitude adjustments during each round showcase its agile 3D maneuvering capabilities. The performance is evaluated by the number of completed round trips within the time limit.

***Hit the Ball.*** The ball is initialized directly above the drone, and the drone hits the ball once to make it land as far as possible. This task is analogous to the typical hitting drill in volleyball and requires both motion control and ball-handling proficiency. In particular, the drone must execute rapid vertical lift, pitch adjustments, and lateral strafing to align precisely with the descending ball, demonstrating another facet of its agile 3D maneuvering. The performance is evaluated by the distance of the ball's landing position from the initial position.

***Solo Bump.*** The ball is initialized directly above the drone, and the drone bumps the ball in place to a specific height as many times as possible within the time limit. This task is analogous to the solo bump drill in human practice and requires motion control, ball-handling proficiency, and stability. During each bump, the drone performs subtle pitch and roll adjustments along with fine vertical thrust modulation to maintain the ball's trajectory, demonstrating its agile 3D maneuvering through precise hover corrections. The performance is evaluated by the number of bumps within the time limit.

## 4.2 Multi-agent cooperative tasks

Multi-agent cooperative tasks are inspired by standard two-player training drills used in volleyball teamwork, including *Bump and Pass*, *Set and Spike (Easy)*, and *Set and Spike (Easy)*. In addition to agile 3D maneuvering, these tasks incorporate turn-based interactions at the intra-team level for coordinated ball-passing sequences.

***Bump and Pass.*** Two drones work together to bump and pass the ball to each other back and forth as many times as possible within the time limit. This task is analogous to the two-player bumping practice in volleyball training and requires homogeneous multi-agent cooperation. The performance is evaluated by the number of successful bumps within the time limit.

***Set and Spike (Easy).*** Two drones take on the role of a setter and an attacker. The setter passes the ball to the attacker, and the attacker then spikes the ball downward to the target region on the opposing side. This task is analogous to the setter-attacker offensive drills in volleyball training and requires heterogeneous multi-agent cooperation. The performance is evaluated by the success rate of the downward spike to the target region.

***Set and Spike (Hard).*** Similar to *Set and Spike (Easy)* task, two drones act as a setter and an attacker to set and spike the ball to the opposing side. The difference is that there is a rule-based defense board on the opposing side to intercept the attacker's spike. The presence of the defense board further improves the difficulty of the task, requiring the drones to optimize their speed, precision,

Table 3: Benchmark result of multi-agent cooperative tasks with different reward settings including without and with shaping reward. *Bump and Pass* is evaluated by the number of bumps, *Set the Spike (Easy)* and *Set the Spike (Hard)* are evaluated by the success rate.

| | Bump and Pass | | Set and Spike (Easy) | | Set and Spike (Hard) | |
| | w.o. shaping | w. shaping | w.o. shaping | w. shaping | w.o. shaping | w. shaping |
|---|---|---|---|---|---|---|
| QMIX | $0.09 \pm 0.01$ | $0.09 \pm 0.00$ | $0.02 \pm 0.00$ | $0.02 \pm 0.00$ | $0.02 \pm 0.00$ | $0.02 \pm 0.00$ |
| MADDPG | $0.79 \pm 0.15$ | $0.84 \pm 0.09$ | $0.22 \pm 0.02$ | $0.23 \pm 0.01$ | $0.22 \pm 0.02$ | $0.22 \pm 0.02$ |
| MAPPO | $\mathbf{11.32 \pm 0.91}$ | $\mathbf{13.71 \pm 0.58}$ | $\mathbf{0.25 \pm 0.00}$ | $\mathbf{0.99 \pm 0.00}$ | $\mathbf{0.25 \pm 0.00}$ | $0.75 \pm 0.01$ |
| HAPPO | $7.95 \pm 3.67$ | $12.14 \pm 0.83$ | $\mathbf{0.25 \pm 0.00}$ | $0.98 \pm 0.00$ | $\mathbf{0.25 \pm 0.00}$ | $0.79 \pm 0.10$ |
| MAT | $7.39 \pm 6.00$ | $13.11 \pm 0.43$ | $\mathbf{0.25 \pm 0.00}$ | $0.89 \pm 0.13$ | $\mathbf{0.25 \pm 0.00}$ | $\mathbf{0.80 \pm 0.11}$ |

and cooperation to defeat the defense board. The performance is evaluated by the success rate of the downward spike that defeats the defense racket.

### 4.3 Multi-agent competitive tasks

Multi-agent competitive tasks follow the standard volleyball match rules, including the competitive *1 vs 1* task and the mixed cooperative-competitive *3 vs 3* and *6 vs 6* tasks. They incorporate competitive and cooperative gameplay, turn-based interaction structure, and agile 3D maneuvering. These tasks demand both the low-level motion control and the high-level strategic play.

*1 vs 1.* Two drones, one positioned on each side of a reduced-size court, compete in a head-to-head volleyball match. A point is scored whenever a drone causes the ball to land in the opponent's court. When the ball is on its side, the drone is allowed only one hit to return the ball to the opponent's court. This two-player zero-sum setting creates a purely competitive environment that requires both precise flight control and strategic gameplay. To evaluate the performance of the learned policy, we consider three typical metrics including the exploitability, the average win rate against other learned policies, and the Elo rating [26]. More specifically, the exploitability is approximated by the gap between the learned best response's win rate against the evaluated policy and its expected win rate at Nash equilibrium, and the Elo rating is computed by running a round-robin tournament between the evaluated policy and a fixed population of policies.

*3 vs 3.* Three drones on each side form a team to compete against the other team on a reduced-size court. During each rally, teammates coordinate to serve, pass, spike and defend, observing the standard limit of three hits per side. This is a challenging mixed cooperative-competitive game that requires both cooperation within the same team and competition between the opposing teams. Moreover, the drones are required to excel at both low-level motion control and high-level game play. We evaluated the policy performance using approximate exploitability, the average win rate against other learned policies, and the Elo rating of the policy.

*6 vs 6.* Six drones per side form teams on a full-size court under the standard three-hits-per-side rule of real-world volleyball. Compared with the *3 vs 3* task, the *6 vs 6* format is substantially more demanding: the larger team size complicates intra-team coordination and role assignment; the full-size court forces drones to cover greater distances and maintain broader defensive coverage; the combinatorial explosion of possible ball trajectories and collision scenarios requires advanced real-time planning and robust collision avoidance; and executing richer tactical schemes necessitates deeper strategic reasoning.

## 5 Benchmark results

We present extensive experiments to benchmark representative (MA)RL and game-theoretic algorithms in our VolleyBots testbed. Specifically, for single-agent tasks, we benchmark five RL algorithms and compare their performance under different action space configurations. For multi-agent cooperative tasks, we evaluate five MARL algorithms and compare their performance with and without reward shaping. For multi-agent competitive tasks, we evaluate four game-theoretic algorithms and provide a comprehensive analysis across multiple evaluation metrics. We identify a key challenge in VolleyBots is the hierarchical decision-making process that requires both low-level motion control and high-level strategic play. We further show the potential of hierarchical policy in

Table 4: Benchmark result of multi-agent competitive tasks including *1 vs 1* and *3 vs 3* with different evaluation metrics.

| | *1 vs 1* | | | *3 vs 3* | | |
|---|---|---|---|---|---|---|
| | Exploitability ↓ | Win Rate ↑ | Elo ↑ | Exploitability ↓ | Win Rate ↑ | Elo ↑ |
| SP | 48.63 | 0.55 | 1072 | **25.76** | 0.59 | 1077 |
| FSP | 30.41 | **0.63** | 927 | 38.86 | 0.52 | 906 |
| $PSRO_{Uniform}$ | 18.51 | 0.35 | 854 | 49.48 | 0.28 | 750 |
| $PSRO_{Nash}$ | **10.74** | 0.47 | **1147** | 35.83 | **0.61** | **1268** |

our VolleyBots testbed by implementing a simple yet effective baseline for the challenging *3 vs 3* task. Detailed discussion about the benchmark algorithms and more experiment results can be found in Appendix E and F.

## 5.1 Results of single-agent tasks

We evaluate five RL algorithms including Deep Q-Network (DQN) [27], Deterministic Policy Gradient (DDPG) [28], Twin Delayed DDPG (TD3) [29], Soft Actor-Critic (SAC) [30], and Proximal Policy Optimization (PPO) [31] in three single-agent tasks. We compare their performance under both CTBR and PRT action spaces. The averaged results over five seeds are shown in Table 2.

For each algorithm, the same set of hyperparameters is used across all tasks to assess its cross-task robustness, while different algorithms are independently tuned for fairness. Details of the hyperparameter tuning process are provided in Appendix F.2.1. Under this setup, PPO consistently outperforms all other methods in every task and under both action-space configurations; by contrast, DQN fails entirely, and DDPG, TD3 and SAC achieve only moderate success. DQN fails because it's limited to discrete actions, forcing coarse binning of continuous drone controls and losing precision. In contrast, while DDPG, TD3, and SAC can handle continuous actions, PPO's clipped surrogate objective and on-policy updates provide greater stability and adaptive exploration, leading to superior performance and stronger cross-task robustness to hyperparameter settings.

Comparing different action spaces, the final results indicate that PRT slightly outperforms CTBR in most tasks. This outcome is likely due to PRT providing more granular control over each motor's thrust, enabling the drone to maximize task-specific performance with precise adjustments. On the other hand, CTBR demonstrates a slightly faster learning speed in some tasks, as its higher-level abstraction simplifies the control process and reduces the learning complexity. For optimal task performance, we use PRT as the default action space in subsequent experiments. Additional experimental results and learning curves are presented in Appendix F.3.

## 5.2 Results of multi-agent cooperative tasks

We evaluate five MARL algorithms including QMIX [32], Multi-Agent DDPG (MADDPG) [33], Multi-Agent PPO (MAPPO) [34], Heterogeneous-Agent PPO (HAPPO) [35], Multi-Agent Transformer (MAT) [36] in three multi-agent cooperative tasks. We also compare their performance with and without reward shaping. The averaged results over five seeds are shown in Table 3.

Comparing the MARL algorithms, on-policy methods like MAPPO, HAPPO, and MAT successfully complete all three cooperative tasks and exhibit comparable performance, while off-policy method like QMIX and MADDPG fails to complete these tasks. These results are consistent with the observation in single-agent experiments, and we use MAPPO as the default algorithm in subsequent experiments for its consistently strong performance and efficiency.

As for different reward functions, it is clear that using reward shaping leads to better performance, especially in more complex tasks like *Set and Spike (Hard)*. This is because the misbehave penalty and task reward alone are usually sparse and make exploration in continuous space challenging. Such sparse setups can serve as benchmarks to evaluate the exploration ability of MARL algorithms. On the other hand, shaping rewards provide intermediate feedback that guides agents toward task-specific objectives more efficiently, and we use shaping rewards in subsequent experiments for efficient learning. More experimental results and learning curves are provided in Appendix F.4.

## 5.3 Results of multi-agent competitive tasks

We evaluate four game-theoretic algorithms: self-play (SP), Fictitious Self-Play (FSP) [37], Policy-Space Response Oracles (PSRO) [38] with a uniform meta-solver (PSRO$_\text{Uniform}$), and a Nash meta-solver (PSRO$_\text{Nash}$) in multi-agent competitive tasks. Algorithms learn effective serving and receiving behaviors in the *1 vs 1* and *3 vs 3* tasks. However, in the most difficult *6 vs 6* task, none of the methods converges to an effective strategy: although the serving drone occasionally hits the ball, it fails to serve the ball to the opponent's court. This finding indicates that the scalability of current algorithms remains limited and requires further improvement. Therefore, we focus our benchmark results on the *1 vs 1* and *3 vs 3* settings. For these two tasks, their performance is evaluated by approximate exploitability, the average win rate against other learned policies, and Elo rating. The results are summarized in Table 4, and head-to-head cross-play win rate heatmaps are shown in Fig. 3. More results and implementation details are provided in Appendix F.5.

In the *1 vs 1* task, all algorithms manage to learn behaviors like reliably returning the ball and maintaining optimal positioning for subsequent volleys. However, the exploitability metric reveals that the learned policies are still far from achieving a Nash equilibrium, indicating limited robustness in this two-player zero-sum game. This performance gap highlights the inherent challenge of hierarchical decision-making in this task, where drones must simultaneously execute precise low-level motion control and engage in high-level strategic gameplay. This challenge presents new opportunities for designing algorithms that can

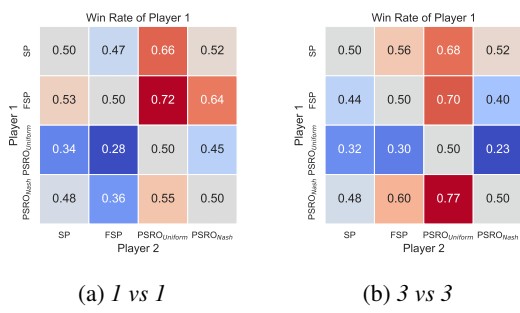

(a) *1 vs 1*  (b) *3 vs 3*

Figure 3: Cross-play heatmap of the *1 vs 1* and *3 vs 3* competitive tasks.

better integrate hierarchical decision-making capabilities. In the *3 vs 3* task, algorithms learn to serve the ball but fail to develop more advanced team strategies. This outcome underscores the compounded challenges in this scenario, where each team of three drones needs to not only cooperate internally but also compete against the opposing team. The increased difficulty of achieving high-level strategic play in such a mixed cooperative-competitive environment further amplifies the hierarchical challenges observed in the *1 vs 1* task.

## 5.4 Hierarchical policy

To illustrate how a simple hierarchical policy might help address the challenges posed by our benchmark, we apply it to the 3 vs 3 task as an example. First, we employ the PPO [31] algorithm to develop a set of low-level drills, including *Hover*, *Serve*, *Pass*, *Set*, and *Attack*. The details of low-level drills can be found in Appendix F.6. Next, we design a rule-based high-level strategic policy that determines when and which low-level drills to assign to each drone. Moreover, for the *Attack*

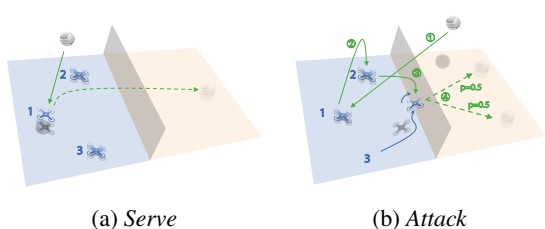

(a) *Serve*  (b) *Attack*

Figure 4: Demonstration of the hierarchical policy selecting *Serve* and *Attack* drills in the *3 vs 3* task.

drill, the high-level policy chooses to hit the ball to the left or right with equal probability. Fig. 4 illustrates two typical demonstrations of the hierarchical policy. In a serving scenario (Fig. 4a), the rule-based high-level strategy assigns the *Serve* drill to drone 1 and the *Hover* drill to the other two drones. In a rally scenario (Fig. 4b), the rule-based strategy assigns the *Pass* drill to drone 1, the *Set* drill to drone 2, and the *Attack* drill to drone 3 sequentially. In accordance with volleyball rules, the high-level policy uses an event-driven mechanism, triggering decisions whenever the ball is hit. As shown in Fig. 3a, the SP policy emerges as the Nash equilibrium among SP, FSP, PSRO$_\text{Uniform}$, and PSRO$_\text{Nash}$ in the 3 vs 3 setting. We evaluate our hierarchical policy against SP over 1,000 episodes and observe a win rate of 69.5%. While the current design of the hierarchical policy is in its early stages, it outperforms the Nash equilibrium baseline, offering valuable inspiration for future developments.

## 6 Sim-to-Real

We use the *Solo Bump* task as a demonstration of the policy's ability to zero-shot transfer to the real world. We use a quadrotor with a rigidly mounted badminton racket. The state of both the drone and the ball is captured using a motion capture system. The drone is modeled as a rigid body, with its position and orientation provided by the motion capture system. The drone's velocity is estimated using an Extended Kalman Filter (EKF) that fuses pose data from the motion capture system and IMU data from the PX4 Autopilot. The ball is modeled as a point mass, with its position sent by the motion capture system and its velocity indirectly computed through a Kalman Filter. The drone's dynamics parameters and the ball's properties are determined through system identification. To simulate real-world noise and imperfect execution of actions, small randomizations are introduced in the ball's initial position, coefficient of restitution, and the ball's rebound velocity after each collision with

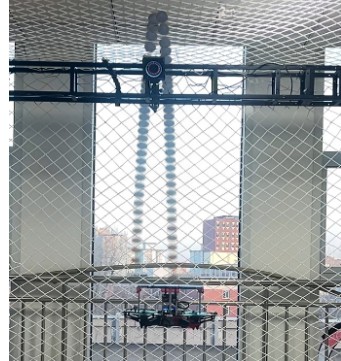

Figure 5: Zero-shot sim-to-real experiment on the *Solo Bump* task.

the drone. Inspired by [39], we also add a smoothness reward to encourage smooth actions. The policy uses CTBR as output and is deployed on the onboard Nvidia Orin processor. As shown in Fig. 5, experiment results show that the drone successfully performs bump tasks multiple times, providing initial evidence of sim-to-real transfer capability. The real-world deployment videos are publicly available on our project website.

## 7 Conclusion

In this work, we introduce VolleyBots, a novel multi-drone volleyball testbed that unifies competitive-cooperative gameplay, turn-based interaction, and agile 3D motion control within a high-fidelity physics simulation. Compounding these features, it demands both motion control and strategic play. Built atop NVIDIA Isaac Sim, VolleyBots offers a structured curriculum of tasks, from single-agent drills and multi-agent cooperative challenges to multi-agent competitive matches. To enable systematic benchmarking, VolleyBots provides implementations of both (MA)RL and game-theoretic baselines across these tasks. Our extensive benchmarks reveal that on-policy RL methods consistently outperform their off-policy counterparts in low-level control tasks, and exhibit stronger cross-task robustness under a single set of hyperparameters. However, both of them still struggle with the tasks demanding both mixed motion control and strategic play, especially in large-scale competitive matches. To address this, we design a simple hierarchical policy that decomposes strategy and control: in the *3 vs 3* task, it achieves a 69.5% win rate against the strongest baseline, highlighting the promise of hierarchical structures. We also showcase the feasibility of deploying policies trained in simulations directly onto physical drones, emphasizing VolleyBots' sim-to-real transfer and practical utility in real-world applications. Going forward, VolleyBots provides a challenging and versatile platform for advancing embodied intelligence in agile robotic systems, inviting novel algorithmic innovations that bridge motion control and strategic play in multi-agent domains.

## Acknowledgments and Disclosure of Funding

We sincerely thank Jiayu Chen, Yuqing Xie, Yinuo Chen, and Sirui Xiang for their valuable discussions, as well as their assistance in experiments and real-world deployment, which have greatly contributed to the development of this work. Their support and collaboration have been instrumental in refining our ideas and improving the quality of this paper.

This research was supported by National Natural Science Foundation of China (No.62406159, 62325405), Postdoctoral Fellowship Program of CPSF under Grant Number (GZC20240830, 2024M761676), China Postdoctoral Science Special Foundation 2024T170496, and Beijing Zhongguancun Academy Project C20250301. Additional support was provided by Tsinghua-Efort Joint Research Center for EAI Computation and Perception, Beijing National Research Center for Information Science and Technology (BNRist), Beijing Innovation Center for Future Chips, and State Key laboratory of Space Network and Communications.

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

# A Impact

This work introduces VolleyBots, a novel robot sports testbed specifically designed to push the boundaries of high-mobility robotic platforms such as drones involving MARL. The broader impacts of this research include advancing the intersection of robotics and MARL, enhancing the decision-making capabilities of drones in complex scenarios. By bridging real-world robotic challenges with MARL, this work aims to inspire future breakthroughs in both robotics and multi-agent AI systems. We do not anticipate any negative societal impacts arising from this work.

# B Limitations

Despite the promising advances of VolleyBots in combining high-level strategic play with low-level motion control, our work has several limitations. First, apart from the *Solo Bump* task, the sim-to-real transferability of the learned policies in other tasks has not yet been evaluated on physical UAV platforms. Second, we rely on fully state-based observations, which overlook challenges such as visual input. Finally, traditional drone control algorithms were not included. Although they struggle with team-level coordination, aggressive maneuvers, and ball interactions, they could still provide informative baselines.

# C Details of VolleyBots environment

## C.1 Court

The volleyball court in our environment is depicted in Fig. 6. The court is divided into two equal halves by the $y$-axis, which serves as the dividing line separating the two teams. The coordinate origin is located at the midpoint of the dividing line, and the $x$-axis extends along the length of the court. The total court length is $18\,m$, with $x = -9$ and $x = 9$ marking the ends of the court. The $y$-axis extends across the width of the court, with a total width of $9\,m$, spanning from $y = -4.5$ to $y = 4.5$. The net is positioned at the center of the court along the $y$-axis, with a height of $2.43\,m$, and spans horizontally from $(0, -4.5)$ to $(0, 4.5)$.

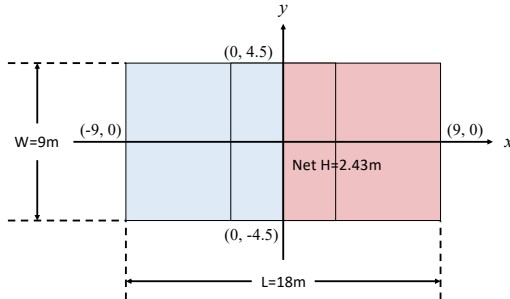

Figure 6: Volleyball court layout in our environment with coordinates.

## C.2 Drone

We use the *Iris* quadrotor model [25] as the primary drone platform, augmented with a virtual "racket" of radius $0.2\,m$ and coefficient of restitution $0.8$ for ball striking. The drone's root state is a vector with dimension 23, including its position, rotation, linear velocity, angular velocity, forward orientation, upward orientation, and normalized rotor speeds.

The control dynamics of a multi-rotor drone are governed by its physical configuration and the interaction of various forces and torques. The system's dynamics can be described as follows:

$$\dot{\boldsymbol{x}}_W = \boldsymbol{v}_W, \quad \dot{\boldsymbol{v}}_W = \boldsymbol{R}_{\boldsymbol{W}\boldsymbol{B}}f + \boldsymbol{g} + \boldsymbol{F} \tag{1}$$

$$\dot{\boldsymbol{q}} = \frac{1}{2}\boldsymbol{q} \otimes \boldsymbol{\omega}, \quad \dot{\boldsymbol{\omega}} = \boldsymbol{J}^{-1}(\boldsymbol{\eta} - \boldsymbol{\omega} \times \boldsymbol{J}\boldsymbol{\omega}) \tag{2}$$

where $x_W$ and $v_W$ represent the position and velocity of the drone in the world frame, $R_{WB}$ is the rotation matrix converting from the body frame to the world frame, $J$ is the diagonal inertia matrix, $g$ denotes gravity, $q$ is the orientation represented by quaternions, and $\omega$ is the angular velocity. The quaternion multiplication operator is denoted by $\otimes$. External forces $F$, including aerodynamic drag and downwash effects, are also considered. The collective thrust $f$ and torque $\eta$ are computed based on per-rotor thrusts $f_i$ as:

$$f = \sum_i R_B^{(i)} f_i \tag{3}$$

$$\eta = \sum_i T_B^{(i)} \times f_i + sign_i k_i f_i \tag{4}$$

where $R_B^{(i)}$ and $T_B^{(i)}$ are the local orientation and translation of the $i$-th rotor in the body frame, $k_i$ is the force-to-moment ratio and $sign_i$ is 1 for clockwise propellers and -1 for counterclockwise propellers.

### C.3 Defense racket

We assume a thin cylindrical racket to mimic a human-held racket for adversarial interactions with a drone. When the ball is hit toward the racket's half of the court, the racket is designed to intercept the ball at a predefined height $h_{pre}$. Since the ball's position and velocity data can be directly acquired, the descent time $t_{pre}$, landing point $p_{ball\_land}$, and pre-collision velocity $v_{ball\_pre}$ can be calculated using projectile motion equations. Additionally, to ensure the ball is returned to a designated position $p_{bdes}$ and crosses the net, the post-collision motion duration $t_{post}$ of the ball is set to a sufficiently large value. This allows the projectile motion equations to similarly determine the post-collision velocity $v_{ball\_post}$. Based on these conditions, the required collision position $p_{collision}$, orientation $\theta_{collision}$ and velocity $v_{collision}$ of the racket can be derived as follows:

$$p_{collision} = p_{ball\_land} \tag{5}$$

$$n_{collision} = \frac{v_{ball\_post} - v_{ball\_pre}}{\|v_{ball\_post} - v_{ball\_pre}\|} = [\sin p \cos r, -\sin r, \cos p \cos r] \tag{6}$$

$$\theta_{collision} = [-\arcsin n_{collision}(2), \arctan \frac{n_{collision}(1)}{n_{collision}(3)}, 0] \tag{7}$$

$$v_{collision} = \frac{1}{1 + \beta}(\beta v_{ball\_pre} + v_{ball\_post}) \tag{8}$$

where $n_{collision}$ represents the normal vector of the racket during impact, $r$ denotes the roll angle of the racket, $p$ denotes the pitch angle, while the yaw angle remains fixed at 0, and $\beta$ represents the restitution coefficient. To simulate the adversarial interaction as realistically as possible, we impose direct constraints on the racket's linear velocity and angular velocity. Based on the simulation time step $t_{step}$ and the descent time $t_{post}$ of the ball, we can calculate the required displacement $d = \frac{p_{bdes} - p_{ball\_land}}{t_{post}} t_{step}$ and rotation angle $\theta = \frac{\theta_{collision}}{t_{post}} t_{step}$ that the racket must achieve within each time step. If both $d$ and $\theta$ do not exceed their respective limits ($d_{max}$ and $\theta_{max}$), the racket moves with linear velocity $d$ and angular velocity $\theta$. Otherwise, the values are set to their corresponding limits $d_{max}$ and $\theta_{max}$.

## D  Details of task design

### D.1  *Back and Forth*

**Task definition.** The drone is initialized at an anchor position $(4.5, 0, 2)$, i.e., the center of the red court with a height of $2\,m$. The other anchor position is $(9.0, 4.5, 2)$, with the target points switching between two designated anchor positions. The drone is required to sprint between two designated anchors to complete as many round trips as possible. 5 steps within a sphere with a $0.6\,m$ radius near the anchor position are required for each stay. The maximum episode length is 800 steps.

Table 5: Reward of single-agent *Back and Forth* task.

| Type | Name | Sparse | Value Range | Description |
|------|------|--------|-------------|-------------|
| Misbehave Penalty | drone_misbehave | ✓ | $\{0, -10\}$ | drone too low or drone too remote |
| Task Reward | dist_to_target | ✗ | $[0, 0.5] \times$ # step | related to drone's distance to the current target |
| | target_stay | ✓ | $\{0, 2.5\} \times$ # in_target | drone stays in the current target region |

Table 6: Reward of single-agent *Hit the Ball* task.

| Type | Name | Sparse | Value Range | Description |
|------|------|--------|-------------|-------------|
| Misbehave Penalty | ball_misbehave | ✓ | $\{0, -10\}$ | ball too low or touch the net or out of court |
| | drone_misbehave | ✓ | $\{0, -10\}$ | drone too low or touches the net |
| | wrong_hit | ✓ | $\{0, -10\}$ | drone does not use the racket to hit the ball |
| Task Reward | success_hit | ✓ | $\{0, 1\}$ | drone hits the ball |
| | distance | ✓ | $[0, +\infty]$ | related to the landing position's distance to the anchor |
| | dist_to_anchor | ✗ | $[-\infty, 0]$ | related to drone's distance to the anchor |

**Observation and reward.** When the action space is Per-Rotor Thrust(PRT), the observation is a vector of dimension 26, which includes the drone's root state and its relative position to the target anchor. When the action space is Collective Thrust and Body Rates (CTBR), the observation dimension is reduced to 22, excluding the drone's throttle. The detailed description of the reward function of this task is listed in Table 5.

**Evaluation metric.** This task is evaluated by the number of target points reached within the time limit. A successful stay is defined as the drone staying 5 steps within a sphere with a $0.6\,m$ radius near the target anchor.

### D.2 *Hit the Ball*

**Task definition.** The drone is initialized randomly around an anchor position $(4.5, 0, 2)$, i.e., the center of the red court with a height of $2\,m$. The drone's initial position is sampled uniformly random from $[4, -0.5, 1.8]$ to $[5, 0.5, 2.2]$. The ball is initialized at $(4.5, 0, 5)$, i.e., $3\,m$ above the anchor position. The ball starts with zero velocity and falls freely. The drone is required to perform a single hit to strike the ball toward the opponent's court, i.e., in the negative direction of the x-axis, aiming for maximum distance. The maximum episode length is 800 steps.

**Observation and reward.** When the action space is Per-Rotor Thrust(PRT), the observation is a vector of dimension 32, which includes the drone's root state, the drone's relative position to the anchor, the ball's relative position to the drone, and the ball's velocity. When the action space is Collective Thrust and Body Rates (CTBR), the observation dimension is reduced to 28, excluding the drone's throttle. The detailed description of the reward function of this task is listed in Table 6.

**Evaluation metric.** This task is evaluated by the distance between the ball's landing position and the anchor position. The ball's landing position is defined as the intersection of its trajectory with the plane $z = 2$.

### D.3 *Solo Bump*

**Task definition.** The drone is initialized randomly around an anchor position $(4.5, 0, 2)$, i.e., the center of the red court with a height of $2\,m$. The drone's initial position is sampled uniformly random from $[4, -0.5, 1.8]$ to $[5, 0.5, 2.2]$. The ball is initialized at $(4.5, 0, 4)$, i.e., $2\,m$ above the anchor position. The ball starts with zero velocity and falls freely. The drone is required to stay within a sphere with $1\,m$ radius near the anchor position and bump the ball as many times as possible. A minimum height of $3.5\,m$ is required for each bump. The maximum episode length is 800 steps.

Table 7: Reward of single-agent *Solo Bump* task.

| Type | Name | Sparse | Value Range | Description |
|---|---|---|---|---|
| Misbehave Penalty | ball_misbehave | ✓ | $\{0, -10\}$ | ball too low or touch the net or out of court |
|  | drone_misbehave | ✓ | $\{0, -10\}$ | drone too low or touches the net |
|  | wrong_hit | ✓ | $\{0, -10\}$ | drone does not use the racket to hit the ball |
| Task Reward | success_hit | ✓ | $\{0, 1\} \times$ # hit | drone hits the ball |
|  | success_height | ✓ | $\{0, 8\} \times$ # hit | ball reaches the minimum height |
| Shaping Reward | dist_to_ball_xy | ✓ | $[0, 1] \times$ # step | related to drone's horizontal distance to the ball |
|  | dist_to_ball_z | ✓ | $[0, 1] \times$ # step | related to drone's clipped vertical distance to the ball |

Table 8: Reward of multi-agent *Bump and Pass* task.

| Type | Name | Sparse | Shared | Value Range | Description |
|---|---|---|---|---|---|
| Misbehave Penalty | ball_misbehave | ✓ | ✓ | $\{0, -10\}$ | ball too low or touches the net or out of court |
|  | drone_misbehave | ✓ | ✗ | $\{0, -10\}$ | drone too low or touches the net |
|  | wrong_hit | ✓ | ✗ | $\{0, -10\}$ | drone hits in the wrong turn |
| Task Reward | success_hit | ✓ | ✓ | $\{0, 1\} \times$ # hit | drone hits the ball |
|  | success_cross | ✓ | ✓ | $\{0, 1\} \times$ # hit | ball crosses the height |
|  | dist_to_anchor | ✗ | ✓ | $[-\infty, 0]$ | related to drone' distance to its anchor |
| Shaping Reward | hit_direction | ✓ | ✗ | $\{0, 1\} \times$ # hit | drone hits the ball towards the other drone |
|  | dist_to_ball | ✗ | ✗ | $[0, 0.05] \times$ # step | related to drone's distance to the ball |

**Observation and reward.** When the action space is Per-Rotor Thrust (PRT), the observation is a vector of dimension 32, which includes the drone's root state, the drone's relative position to the anchor, the ball's relative position to the drone, and the ball's velocity. When the action space is Collective Thrust and Body Rates (CTBR), the observation dimension is reduced to 28, excluding the drone's throttle. The detailed description of the reward function of this task is listed in Table 7.

**Evaluation metric.** This task is evaluated by the number of successful consecutive bumps performed by the drone. A successful bump is defined as the drone hitting the ball such that the ball's highest height exceeds $3.5\,m$ but not exceeds $4.5\,m$.

### D.4 *Bump and Pass*

**Task definition.** Drone 1 is initialized randomly around anchor 1 with position $(4.5, -2.5, 2)$, and Drone 2 is initialized randomly around anchor 2 with position $(4.5, 2.5, 2)$. The initial position of drone 1 is sampled uniformly random from $(4, -3, 1.8)$ to $(5, -2, 2.2)$, and the initial position of drone 2 is sampled uniformly random from $(4, 2, 1.8)$ to $(5, 3, 2.2)$. The ball is initialized at $(4.5, -2.5, 4)$, i.e., $2\,m$ above anchor 1. The ball starts with zero velocity and falls freely. The drones are required to stay within a sphere with $0.5\,m$ radius near their anchors and bump the ball to pass it to each other in turns as many times as possible. A minimum height of $4\,m$ is required for each bump. The maximum episode length is 800 steps.

**Observation and reward.** The drone's observation is a vector of dimension 39 including the drone's root state, the drone's relative position to the anchor, the drone's id, the current turn (which drone should hit the ball), the ball's relative position to the drone, the ball's velocity, and the other drone's relative position to the drone. The detailed description of the reward function of this task is listed in Table 8.

**Evaluation metric.** This task is evaluated by the number of successful consecutive bumps performed by the drones. A successful bump is defined as the drone hitting the ball such that the ball's highest height exceeds $4\,m$ and lands near the other drone.

Table 9: Reward of multi-agent *Set and Spike (Easy)* task.

| Type | Name | Sparse | Shared | Value Range | Description |
|------|------|--------|--------|-------------|-------------|
| Misbehave Penalty | ball_misbehave | ✓ | ✓ | $\{0, -10\}$ | ball too low or touches the net or out of court |
| | drone_misbehave | ✓ | ✗ | $\{0, -10\}$ | drone too low or touches the net |
| | wrong_hit | ✓ | ✗ | $\{0, -10\}$ | drone hits in the wrong turn |
| Task Reward | success_hit | ✓ | ✓ | $\{0, 5\} \times$ # hit | drone hits the ball |
| | downward_spike | ✓ | ✓ | $\{0, 5\} \times$ # spike | ball's velocity is downward after spike |
| | success_cross | ✓ | ✓ | $\{0, 5\}$ | ball crosses the net |
| | in_target | ✓ | ✓ | $\{0, 5\}$ | ball lands in the target region |
| | dist_to_anchor | ✗ | ✓ | $[-\infty, 0]$ | related to drone's distance its anchor |
| Shaping Reward | hit_direction | ✓ | ✗ | $\{0, 1\} \times$ # hit | drone hits the ball towards its target |
| | spike_velociy | ✓ | ✓ | $[0, +\infty] \times$ # spike | related to ball's downward velocity after spike |
| | dist_to_ball | ✗ | ✗ | $[0, 0.05] \times$ # step | related to drone's distance to the ball |
| | dist_to_target | ✗ | ✓ | $[0, 2]$ | related to ball's landing position to the target |

Table 10: Reward of multi-agent *Set and Spike (Hard)* task.

| Type | Name | Sparse | Shared | Value Range | Description |
|------|------|--------|--------|-------------|-------------|
| Misbehave Penalty | ball_misbehave | ✓ | ✓ | $\{0, -10\}$ | ball too low or touches the net or out of court |
| | drone_misbehave | ✓ | ✗ | $\{0, -10\}$ | drone too low or touches the net |
| | wrong_hit | ✓ | ✗ | $\{0, -10\}$ | drone hits in the wrong turn |
| Task Reward | success_hit | ✓ | ✓ | $\{0, 5\} \times$ # hit | drone hits the ball |
| | downward_spike | ✓ | ✓ | $\{0, 5\} \times$ # spike | ball's velocity is downward after spike |
| | success_cross | ✓ | ✓ | $\{0, 5\}$ | ball crosses the net |
| | success_spike | ✓ | ✓ | $\{0, 5\}$ | defense racket fails to intercept |
| | dist_to_anchor | ✗ | ✓ | $[-\infty, 0]$ | related to drone's distance to its anchor |
| Shaping Reward | hit_direction | ✓ | ✗ | $\{0, 1\} \times$ # hit | drone hits the ball towards their targets |
| | spike_velociy | ✓ | ✓ | $[0, +\infty] \times$ # spike | related to ball's downward velocity after spike |
| | dist_to_ball | ✗ | ✗ | $[0, 0.05] \times$ # step | related to drone's distance to the ball |

### D.5 *Set and Spike (Easy)*

**Task definition.** Drone 1 (setter) is initialized randomly around anchor 1 with position $(2, -2.5, 2.5)$, and Drone 2 (attacker) is initialized randomly around anchor 2 with position $(2, 2.5, 3.5)$. The initial position of drone 1 is sampled uniformly random from $(1.5, -3, 2.3)$ to $(2.5, -2, 2.7)$, and the initial position of drone 2 is sampled uniformly random from $(1.5, 2, 3.3)$ to $(2.5, 3, 3.7)$. The ball is initialized at $(2, -2.5, 4.5)$, i.e., $2\,m$ above anchor 1. The ball starts with zero velocity and falls freely. The drones are required to stay within a sphere with $0.5\,m$ radius near their anchors. The setter is required to pass the ball to the attacker, and the attacker then spikes the ball downward to the target region in the opposing side. The target region is a circular area on the ground, centered at $(4.5, 0)$ with a radius of $1\,m$. The maximum episode length is 800 steps.

**Observation and reward.** The drone's observation is a vector of dimension $40$ including the drone's root state, the drone's relative position to the anchor, the drone's id, the current turn (how many times the ball has been hit), the ball's relative position to the drone, the ball's velocity, and the other drone's relative position to the drone. The detailed description of the reward function of this task is listed in Table 9.

**Evaluation metric.** This task is evaluated by the success rate of set and spike. A successful set and spike consist of four parts, (1) setter_hit: the setter hits the ball; (2) attacker_hit: the attacker hits the ball; (3) downward_spike: the velocity of the ball after the attacker hit is downward, i.e., $v_z < 0$; (4) in_target: the ball's landing position is within the target region. The success rate is computed as $1/4 \times$ (setter_hit + attacker_hit + downward_spike + in_target).

### D.6 *Set and Spike (Hard)*

**Task definition.** Drone 1 (setter) is initialized randomly around anchor 1 with position $(2, -2.5, 2.5)$, and Drone 2 (attacker) is initialized randomly around anchor 2 with position $(2, 2.5, 3.5)$. The initial position of drone 1 is sampled uniformly random from $(1.5, -3, 2.3)$ to

Table 11: Reward of multi-agent *1 vs 1* task.

| Type | Name | Sparse | Shared | Value Range | Description |
|---|---|---|---|---|---|
| Misbehave Penalty | drone_misbehave | ✓ | ✗ | $\{0, -100\}$ | drone too low or touches the net |
| | drone_out_of_court | ✗ | ✗ | $[0, 0.2] \times$ # step | related to drone's distance out of its court |
| Task Reward | win_or_lose | ✓ | ✗ | $\{-100, 0, 100\}$ | drone wins or loses the game |
| Shaping Reward | success_hit | ✓ | ✗ | $\{0, 5\} \times$ # hit | drone hits the ball |
| | dist_to_ball | ✗ | ✗ | $[0, 0.5] \times$ # step | related to drone's distance to the ball |

Table 12: Reward of multi-agent *3 vs 3* task.

| Type | Name | Sparse | Shared | Value Range | Description |
|---|---|---|---|---|---|
| Misbehave Penalty | drone_misbehave | ✓ | ✗ | $\{0, -100\}$ | drone too low or touches the net. |
| | drone_collision | ✓ | ✗ | $\{0, -100\}$ | drone collides with its teammate. |
| Task Reward | win_or_lose | ✓ | ✓ | $\{-100, 0, 100\}$ | drones win or lose the game |
| Shaping Reward | success_hit | ✓ | ✓ | $\{0, 10\} \times$ # hit | drone hits the ball |
| | dist_to_anchor | ✗ | ✗ | $[0, 0.05] \times$ # step | related to drone's distance to its anchor |
| | dist_to_ball | ✗ | ✗ | $[0, 0.5] \times$ # step | related to drone's distance to the ball |

$(2.5, -2, 2.7)$, and the initial position of drone 2 is sampled uniformly random from $(1.5, 2, 3.3)$ to $(2.5, 3, 3.7)$. The ball is initialized at $(2, -2.5, 4.5)$, i.e., $2\,m$ above anchor 1. The ball starts with zero velocity and falls freely. The racket is initialized at $(-4, 0, 0.5)$, i.e., the center of the opposing side. The drones are required to stay within a sphere with $0.5\,m$ radius near their anchors. The setter is required to pass the ball to the attacker, and the attacker then spikes the ball downward to the opponent's court without being intercepted by the defense racket. The maximum episode length is 800 steps.

**Observation and reward.** The drone's observation is a vector of dimension 40 including the drone's root state, the drone's relative position to the anchor, the drone's id, the current turn (how many times the ball has been hit), the ball's relative position to the drone, the ball's velocity, and the other drone's relative position to the drone. The detailed description of the reward function of this task is listed in Table 10.

**Evaluation metric.** This task is evaluated by the success rate of set and spike. A successful set and spike consist of four parts, (1) setter_hit: the setter hits the ball; (2) attacker_hit: the attacker hits the ball; (3) downward_spike: the velocity of the ball after the attacker hit is downward, i.e., $v_z < 0$; (4) success_spike: the ball's landing position is within the opponent's court without being intercepted by the defense racket. The success rate is computed as $1/4 \times$ (setter_hit + attacker_hit + downward_spike + success_spike).

### D.7   *1 vs 1*

**Task definition.** Two drones are required to play 1 vs 1 volleyball in a reduced-size court of $6\,m \times 3\,m$. Drone 1 is initialized randomly around anchor 1 with position $(1.5, 0.0, 2.0)$, i.e., the center of the red court with height $2\,m$, and Drone 2 is initialized randomly around anchor 2 with position $(-1.5, 0.0, 2.0)$, i.e., the center of the blue court with height $2\,m$. The initial position of drone 1 is sampled uniformly random from $(1.4, -0.1, 1.9)$ to $(1.6, 0.1, 2.1)$, and the initial position of drone 2 is sampled uniformly random from $(-1.4, -0.1, 1.9)$ to $(-1.6, 0.1, 2.1)$. At the start of a game (i.e. an episode), one of the two drones is randomly chosen to serve the ball, which is initialized $1.5\,m$ above the drone. The ball starts with zero velocity and falls freely. The game ends when one of the drones wins the game or one of the drones crashes. The maximum episode length is 800 steps.

**Observation and reward.** The drone's observation is a vector of dimension 39 including the drone's root state, the drone's relative position to the anchor, the drone's id, the current turn (which drone should hit the ball), the ball's relative position to the drone, the ball's velocity, and the other drone's relative position to the drone. The detailed description of the reward function of this task is listed in Table 11.

Table 13: Reward of multi-agent *6 vs 6* task.

| Type | Name | Sparse | Shared | Value Range | Description |
|---|---|---|---|---|---|
| Misbehave Penalty | drone_misbehave | ✓ | ✗ | $\{0, -100\}$ | drone too low or touches the net. |
| | drone_collision | ✓ | ✗ | $\{0, -100\}$ | drone collides with its teammate. |
| Task Reward | win_or_lose | ✓ | ✓ | $\{-100, 0, 100\}$ | drones win or lose the game |
| Shaping Reward | success_hit | ✓ | ✓ | $\{0, 10\} \times$ # hit | drone hits the ball |
| | dist_to_anchor | ✗ | ✗ | $[0, 0.05] \times$ # step | related to drone's distance to its anchor |
| | dist_to_ball | ✗ | ✗ | $[0, 0.5] \times$ # step | related to drone's distance to the ball |

**Evaluation metric.** The drone wins the game by landing the ball in the opponent's court or causing the opponent to commit a violation. These violations include (1) crossing the net, (2) hitting the ball on the wrong turn, (3) hitting the ball with part of the drone body instead of the racket, (4) hitting the ball out of court, and (5) hitting the ball into the net.

To comprehensively evaluate the performance of strategies in the *1 vs 1* task, we consider three evaluation metrics: exploitability, win rate, and Elo rating. These metrics provide complementary insights into the quality and robustness of the learned policies.

- **Exploitability:** Exploitability is a fundamental measure of how close a strategy is to a Nash equilibrium. It is defined as the difference between the payoff of a best response (BR) against the strategy and the payoff of the strategy itself. Mathematically, for a strategy $\pi$, the exploitability is given by:

$$\text{Exploitability}(\pi) = \max_{\pi'} U(\pi', \pi) - U(\pi, \pi),$$

  where $U(\pi_1, \pi_2)$ represents the utility obtained by $\pi_1$ when playing against $\pi_2$. The meaning of exploitability is that smaller values indicate a strategy closer to Nash equilibrium, where it becomes increasingly difficult to exploit. Since exact computation of exploitability is often infeasible in real-world tasks, we instead use approximate exploitability. In this task, we fix the strategy on one side and train an approximate best response on the other side to maximize its utility, i.e., win rate. The difference between the BR's win rate and the evaluated policy's win rate then serves as the approximate exploitability.

- **Win rate:** Since exact exploitability is challenging to compute, a practical alternative is to evaluate the win rate through cross-play with other learned policy populations. Specifically, we compute the average win rate of the evaluated policy when matched against other learned policies. Higher average win rates typically suggest stronger strategies. However, due to the transitive nature of zero-sum games [40], a high win rate against specific opponent populations does not necessarily imply overall mastery of the game. Thus, while win rate is a useful reference metric, it cannot be the sole criterion for assessing strategy strength.

- **Elo rating:** Elo rating is a widely used metric for evaluating the relative strength of strategies within a population. It is computed based on head-to-head match results, where the expected win probability between two strategies is determined by their Elo difference. After each match, the Elo ratings of the strategies are updated based on the match outcome. While a higher Elo rating indicates better performance within the given population, it does not necessarily imply proximity to Nash equilibrium. A strategy with a higher Elo might simply be more effective against the specific population, rather than being universally robust. Therefore, Elo complements exploitability by capturing population-specific relative performance.

### D.8  *3 vs 3*

**Task definition.** The task involves two teams of drones competing in a 3 vs 3 volleyball match within a reduced-size court of $9\,m \times 4.5\,m$. Drone 1, Drone 2, and Drone 3 belong to *Team 1* and are initialized at positions $(3.0, -1.5, 2.0)$, $(3.0, 1.5, 2.0)$ and $(6.0, 0.0, 2.0)$ respectively. Similarly, Drone 4, Drone 5, and Drone 6 belong to *Team 2* and are initialized at positions $(-3.0, -1.5, 2.0)$, $(-3.0, 1.5, 2.0)$ and $(-6.0, 0.0, 2.0)$ respectively. At the start of a game (i.e., an episode), one of the two teams is randomly selected to serve the ball. The ball is initialized at a position $3\,m$ directly

above the serving drone. The ball starts with zero velocity and falls freely. The game ends when one of the teams wins the game or one of the drones crashes. The maximum episode length is 500 steps.

**Observation and reward.** The drone's observation is a vector of dimension 57 including the drone's root state, the drone's relative position to the anchor, the ball's relative position to the drone, the ball's velocity, the current turn (which team should hit the ball), the drone's id, a flag indicating whether the drone is allowed to hit the ball, and the other drone's positions. The detailed description of the reward function of this task is listed in Table 12.

**Evaluation metric.** Similar to the *1 vs 1* task, either of the two teams wins the game by landing the ball in the opponent's court or causing the opponent to commit a violation. These violations include (1) crossing the net, (2) hitting the ball on the wrong turn, (3) hitting the ball with part of the drone body, rather than the racket, (4) hitting the ball out of court, and (5) hitting the ball into the net. The task performance is also evaluated by the three metric metrics including exploitability, win rate, and Elo as described in the *1 vs 1* task.

### D.9   *6 vs 6*

**Task definition.** The task involves two teams of drones competing in a 6 vs 6 volleyball match within a standard-size court of $12\,m \times 6\,m$. Drone 1 to Drone 6 belong to *Team 1* and are initialized at positions $(3.0, -3.0, 2.0)$, $(3.0, 0.0, 2.0)$, $(3.0, 3.0, 2.0)$, $(6.0, -3.0, 2.0)$, $(9.0, 0.0, 2.0)$ and $(6.0, 3.0, 2.0)$ respectively. Similarly, Drone 7 to Drone 12 belong to *Team 2* and are initialized at positions $(-3.0, 3.0, 2.0)$, $(-3.0, 0.0, 2.0)$, $(-3.0, -3.0, 2.0)$, $(-6.0, 3.0, 2.0)$, $(-9.0, 0.0, 2.0)$ and $(-6.0, -3.0, 2.0)$ respectively. At the start of a game (i.e., an episode), one of the two teams is randomly selected to serve the ball. The ball is initialized at a position $3\,m$ directly above the serving drone. The ball starts with zero velocity and falls freely. The game ends when one of the teams wins the game or one of the drones crashes. The maximum episode length is 500 steps.

**Observation and reward.** The drone's observation is a vector of dimension 78 including the drone's root state, the drone's relative position to the anchor, the ball's relative position to the drone, the ball's velocity, the current turn (which team should hit the ball), the drone's id, a flag indicating whether the drone is allowed to hit the ball, and the other drone's positions. The detailed description of the reward function of this task is listed in Table 13.

**Evaluation metric.** Either of the two teams wins the game by landing the ball in the opponent's court or causing the opponent to commit a violation. These violations include (1) crossing the net, (2) hitting the ball on the wrong turn, (3) hitting the ball with part of the drone body, rather than the racket, (4) hitting the ball out of court, and (5) hitting the ball into the net.

## E   Discussion of benchmark algorithms

### E.1   Reinforcement learning algorithms

To explore the capabilities of our testbed while also providing baseline results, we implement and benchmark a spectrum of popular RL and game-theoretic algorithms on the proposed tasks.

**Single-agent RL.** In single-agent scenarios, we consider five commonly used algorithms. Deep Q-Network (DQN) [27] is a value-based, off-policy method that approximates action-value functions for discrete action spaces using experience replay and a target network to stabilize learning. Deep Deterministic Policy Gradient (DDPG) [28] is an off-policy actor-critic approach relying on a deterministic policy and an experience replay buffer to handle continuous actions. Twin Delayed DDPG (TD3) [29] builds on DDPG by employing two Q-networks to mitigate overestimation bias, delaying policy updates, and adding target policy smoothing for improved stability. Soft Actor-Critic (SAC) [30] is an off-policy actor-critic algorithm that maximizes a combined reward-and-entropy objective, promoting robust exploration via a maximum-entropy framework. Proximal Policy Optimization (PPO) [31] adopts a clipped objective to stabilize on-policy learning updates by constraining policy changes. Overall, these methods provide contrasting paradigms for tackling single-agent continuous tasks.

Table 14: Shared hyperparameters used for DQN, DDPG, TD3, and SAC in single-agent tasks.

| hyperparameters | value | hyperparameters | value | hyperparameters | value |
|---|---|---|---|---|---|
| optimizer | Adam | max grad norm | 10 | lr | $5 \times 10^{-4}$ |
| buffer length | 64 | buffer size | $1 \times 10^{6}$ | batch size | 4096 |
| gamma | 0.95 | tau | 0.005 | target update interval | 4 |
| max episode length | 800 | num envs | 4096 | train steps | $5 \times 10^{8}$ |

Table 15: Algorithm-specific hyperparameters used for DQN, DDPG, TD3, and SAC in single-agent tasks.

| Algorithms | DQN | DDPG | TD3 | SAC |
|---|---|---|---|---|
| actor network | / | MLP | MLP | MLP |
| critic network | MLP | MLP | MLP | MLP |
| MLP hidden sizes | $[256, 128]$ | $[256, 128, 128]$ | $[256, 128, 128]$ | $[256, 128, 128]$ |
| critic loss | / | smooth L1 | smooth L1 | smooth L1 |
| discrete bin | 2 | / | / | / |

**Multi-agent RL.** For tasks with multiple drones, we evaluate five representative multi-agent algorithms. QMIX [32] is a value-based method that factorizes the global action-value function into individual agent utilities via a monotonic mixing network, enabling centralized training with decentralized execution. Multi-Agent DDPG (MADDPG) [33] extends DDPG with a centralized critic for each agent, while policies remain decentralized. Multi-Agent PPO (MAPPO) [34] incorporates a shared value function to improve both coordination and sample efficiency. Heterogeneous-Agent PPO (HAPPO) [35] adapts PPO techniques to handle distinct roles or capabilities among agents. Multi-Agent Transformer (MAT) [36] leverages a transformer-based architecture to enable attention-driven collaboration. Taken together, these algorithms offer a diverse set of baselines for multi-agent cooperation.

**Game-theoretic algorithms.** For multi-agent competitive tasks, we consider several representative game-theoretic algorithms in the literature [41]. Self-play (SP) trains agents against the current version of themselves, allowing a single policy to evolve efficiently. Fictitious Self-Play (FSP) [37] trains agents against the average policy by maintaining a pool of past checkpoints. Policy-Space Response Oracles (PSRO) [38] iteratively add the best responses to the mixture of a growing policy population. The mixture policy is determined by a meta-solver. $\text{PSRO}_{\text{uniform}}$ uses a uniform meta-solver that samples policies with equal probability, while $\text{PSRO}_{\text{Nash}}$ uses a Nash meta-solver that samples policies according to the Nash equilibrium. These methods provide an extensive benchmark for game-theoretic algorithms in multi-agent competition with both motion control and strategic play. There are also some algorithms like Team-PSRO [42] and Fictitious Cross-Play (FXP) [43] that are designed specifically for mixed cooperative-competitive games and can be integrated in our testbed in future work.

# F    Details of benchmark experiments

## F.1    Experimental Platform and Computational Resources

Our experiments were conducted on a workstation equipped with NVIDIA GeForce RTX 4090 or RTX 3090 GPUs, 128 GB of RAM, and Ubuntu 20.04 LTS. The software environment included CUDA 12.4, Python 3.10, PyTorch 2.0, and NVIDIA Isaac Sim 2023.1.0. All single-agent and multi-agent cooperative experiments completed in under 12 hours, while multi-agent competitive experiments finished in under 24 hours.

Table 16: Shared hyperparameters used for (MA)PPO, HAPPO, and MAT in single-agent tasks and the multi-agent tasks.

| hyperparameters | value | hyperparameters | value | hyperparameters | value |
|---|---|---|---|---|---|
| optimizer | Adam | max grad norm | 10 | entropy coef | 0.001 |
| buffer length | 64 | num minibatches | 16 | ppo epochs | 4 |
| value norm | ValueNorm1 | clip param | 0.1 | normalize advantages | True |
| use huber loss | True | huber delta | 10 | gae lambda | 0.95 |
| use orthogonal | True | gain | 0.01 | gae gamma | 0.995 |
| max episode length | 800 | num envs | 4096 | train steps | $1 \times 10^9$ |

Table 17: Algorithm-specific hyperparameters used for (MA)PPO, HAPPO, and MAT in the single-agent tasks and multi-agent tasks.

| Algorithms | (MA)PPO | HAPPO | MAT |
|---|---|---|---|
| actor lr | $5 \times 10^{-4}$ | $5 \times 10^{-4}$ | $3 \times 10^{-5}$ |
| critic lr | $5 \times 10^{-4}$ | $5 \times 10^{-4}$ | $3 \times 10^{-5}$ |
| share actor | True | False | / |
| hidden sizes | $[256, 128, 128]$ | $[256, 128]$ | $[256, 256, 256]$ |
| num blocks | / | / | 3 |
| num head | / | / | 8 |

## F.2 Hyperparameters of benchmarking algorithms

### F.2.1 Single-agent tasks.

In the single-agent setting, we tune hyperparameters on a simpler task, *Hover*, proposed in OmniDrones [24]. In this task, the drone starts from a randomized position and heading, moves toward a randomized target, and then maintains a stable pose without drift. The reward function depends on position error, heading alignment, uprightness, and angular stability. We perform random search over the hyperparameter space, and the best configurations found on *Hover* yield the baseline performance shown in Table 19. On this simple task, DDPG, TD3, SAC, and PPO achieve comparable performance, while DQN fails. For fairness and reproducibility, we fix each algorithm's configuration obtained on *Hover* and apply it unchanged to all other single-agent tasks reported in Table 2. This setup allows us to evaluate cross-task robustness without task-specific tuning.

The hyperparameters adopted for DQN, DDPG, TD3, and SAC in the single-agent tasks are listed in Table 14 and 15. The hyperparameters adopted for PPO in the single-agent tasks are listed in Table 16 and 17. All algorithms are trained for $5 \times 10^8$ environment steps in each task.

### F.2.2 Multi-agent cooperative tasks

The hyperparameters adopted for different algorithms in multi-agent cooperative tasks are listed in Table 16, 17, and 18. All algorithms are trained for $1 \times 10^9$ environment steps in each task.

### F.2.3 Multi-agent competitive tasks.

**Training.** For self-play (SP) in *1 vs 1*, *3 vs 3*, and *6 vs 6* competitive tasks, we adopt the MAPPO algorithm with shared actor networks and shared critic networks between two teams, in order to make sure two teams utilize the same policy. Also, we transform the samples from both sides into symmetric ones and then use these symmetric samples to update the network together. The hyperparameters employed here are the same as those used in the MAPPO algorithm for multi-agent cooperative tasks.

The PSRO algorithm for *1 vs 1* competitive task instantiates a PPO agent for training one of the two drones while the other drone maintains a fixed policy. Similarly, the PSRO algorithm for the *3 vs 3* and *6 vs 6* tasks assigns each team to be controlled by MAPPO. We adopt the same set of hyperparameters listed in Table 16 and 17 for the (MA)PPO agent. In each iteration, the (MA)PPO agent is trained against the current population. Here, we offer two versions of meta-strategy solver,

Table 18: Hyperparameters used for QMIX in multi-agent tasks.

| hyperparameters | value | hyperparameters | value | hyperparameters | value |
|---|---|---|---|---|---|
| optimizer | Adam | q_net and q_mixer network | MLP | MLP hidden sizes | $[256, 128]$ |
| lr | $5 \times 10^{-4}$ | buffer length | 64 | buffer size | 1024 |
| batch size | 128 | gamma | 0.99 | max grad norm | 10 |
| discrete bin | 2 | tau | 0.005 | target update interval | 4 |
| max episode length | 800 | num envs | 4096 | train steps | $1 \times 10^9$ |

Table 19: Results of hyperparameter tuning for single-agent RL algorithms on the *Hover* task.

| Algorithms | DDPG | TD3 | SAC | PPO |
|---|---|---|---|---|
| Return | $1168.01 \pm 16.83$ | $1212.82 \pm 8.81$ | $1249.39 \pm 4.55$ | $1196.68 \pm 3.01$ |

$\text{PSRO}_{\text{Uniform}}$ and $\text{PSRO}_{\text{Nash}}$. Training is considered converged when the agent achieves over 90% win rate with a standard deviation below 0.05. The iteration ends when the agent reaches convergence or reaches a maximum of iteration steps of 5000. The trained actor is then added to the population for the next iteration.

For Fictitious Self-Play (FSP) in competitive tasks, we slightly modify $\text{PSRO}_{\text{Uniform}}$ so that in each iteration, the (MA)PPO agent inherits the learned policy from the previous iteration as initialization. Naturally, other hyperparameters and settings remain the same for a fair comparison.

The algorithm leverages 2048 parallel environments for the *1 vs 1* and *3 vs 3* tasks, and 800 parallel environments for the *6 vs 6* task. In this work, we report the results of different algorithms given a total budget of $1 \times 10^9$ environmental steps.

**Evaluation.** The evaluation of exploitability requires evaluating the payoff of the best response (BR) over the trained policy or population from different algorithms. Here, we approximate the BR to each policy or population by learning an additional RL agent against the trained policy or population. In practice, this is done by performing an additional iteration of PSRO, where the opponent is fixed as the trained policy/population. In order to approximate the ideal BR as closely as possible, we initialize the BR policy with the latest FSP policy, given that FSP yields the best empirical performance in our experiments. We train the BR policy for 5000 training step with 2048 parallel environments. We disable the convergence condition for early termination and report the evaluated win rate to calculate the approximate exploitability. Importantly, to approximate the BR of the trained SP policy in the *3 vs 3* task, we employ two distinct BR policies for the serve and rally scenarios, respectively. For the BR to serve, we directly use the latest FSP policy without further training, while for the BR to rally, we train a dedicated policy against the SP policy. The overall win rate of this BR is then computed as the average win rate across these two scenarios, given that each side has an equal serve probability.

We run 1,000 games for each pair of policies to generate the cross-play win rate heatmap, covering 6 matchup scenarios, resulting in a total of 6,000 games. In each game, both policies are sampled from their respective policy populations based on the meta-strategy and play until a winner is determined.

Moreover, we use an open-source Elo implementation [44]. The coefficient $K$ is set to 168, and the initial Elo rating for all policies is 1000. We conduct 12000 games among four policies. The number of games played between any two policies is guaranteed to be the same. Specifically, in each round, 6 different matchups are played. Each policy participates in 3 matchups, competing against different opponent policies. A total of 2000 rounds are carried out, amounting to 12000 games in total. The game results are sampled and generated based on the cross-play results.

### F.3 Results of single-agent tasks

Fig. 7 plots the training progress of five single-agent algorithms on three single-agent volleyball tasks under both CTBR (top row) and PRT (bottom row) action space, averaged over five seeds. Across every task, PPO (orange) converges fastest and to the highest performance, stabilizing at roughly $9 - 10$ successful reaches in *Back and Forth*, $10 - 11$m in *Hit the Ball*, and $8 - 12$ bumps in *Solo*

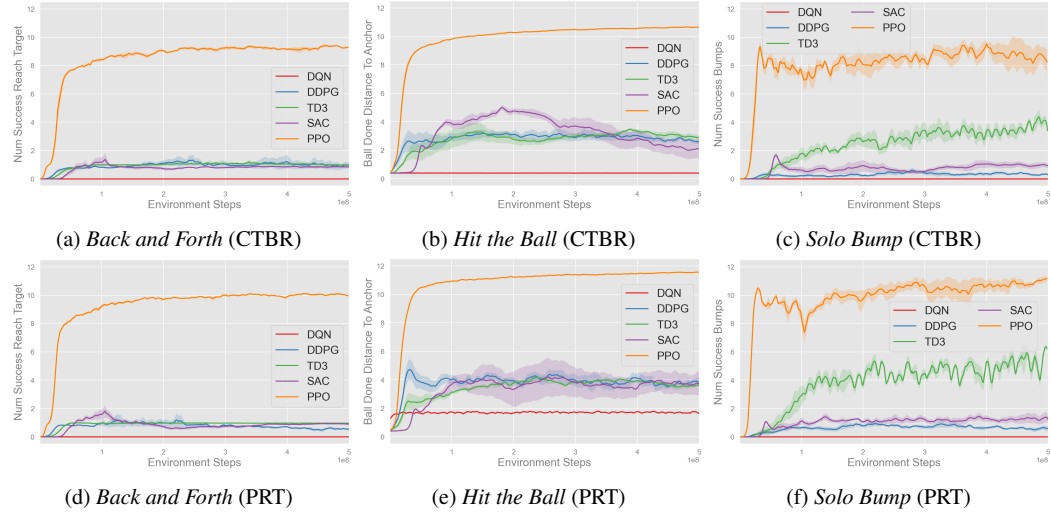

Figure 7: Training curves of single-agent tasks over five seeds.

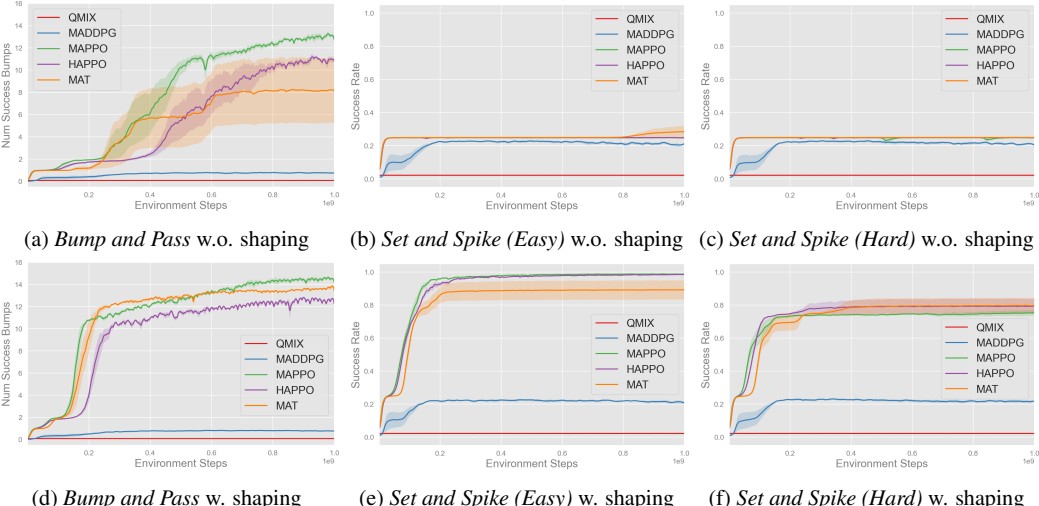

Figure 8: Training curves of multi-agent cooperative tasks over five seeds.

*Bump*. TD3 (green), SAC (purple), and DDPG (blue) exhibit comparable moderate performance across most tasks, with TD3 notably outperforming SAC and DDPG in *Solo Bump*. In contrast, DQN (red) fails to make meaningful progress in any of the tasks. Moreover, each algorithm exhibits comparable behavior under both CTBR and PRT action spaces, with slightly better final performance under PRT for most methods and tasks.

### F.4 Results of multi-agent cooperative tasks

The training curves of different algorithms in multi-agent tasks are shown in Fig. 8. MAPPO (green), HAPPO (purple) and MAT (orange) achieve the strongest overall performance. In the *Bump and Pass* task without reward shaping, MAPPO learns fastest and attains the highest number of successful bumps, outperforming both HAPPO and MAT. By contrast, MADDPG (blue) delivers only modest gains, struggling particularly in *Bump and Pass*, and QMIX (red) fails to make meaningful progress in any of the tasks.

Additionally, we can observe that the presence of shaping rewards has a significant impact on task results. Adding shaping rewards clearly improves the performance and accelerates the learning process. In *Bump and Pass*, the task learns more slowly without shaping rewards because the policy

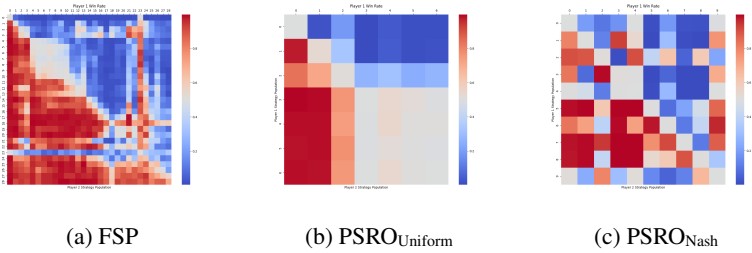

| (a) FSP | (b) PSRO$_{\text{Uniform}}$ | (c) PSRO$_{\text{Nash}}$ |

Figure 9: Win rate heatmaps of the population in the *1 vs 1* task.

must explore which direction to hit the ball, requiring many more steps. The hit direction reward in shaping rewards accelerates this process. In *Set and Spike (Easy)* and *Set and Spike (Hard)*, all algorithms without shaping rewards have a success rate of only $0.25$ because they only learn to make the setter hit the ball, but not toward the hitter. As a result, the attacker fails to hit the ball. The hit direction reward in shaping rewards helps accelerate this process.

## F.5    Results of multi-agent competitive tasks

We provide a more detailed win rate evaluation of the PSRO populations from the *1 vs 1* task in Fig. 9, where each policy in the PSRO population is evaluated against all other policies. In these heatmaps, the ordinate and abscissa represent the policy for drone 1 and drone 2 respectively. The heat of cells represents the evaluated win rate of drone 1, i.e. red means a higher win rate and blue means a lower win rate. Intuitively, each row represents a policy's performance against each policy of the population while playing as drone 1. A red cell indicates that the drone 1 policy outperforms the specific drone 2 policy. A full red row means that the policy outperforms all other policies.

Evidently, FSP attains more iterations than PSRO$_{\text{Uniform}}$ and PSRO$_{\text{Nash}}$ given a budget of $1 \times 10^9$ steps, which yields a faster convergence speed. This advantage comes from the fact that FSP inherits the learned policy from the previous iteration, which serves as an advantageous initialization for the current iteration. In contrast, PSRO$_{\text{Uniform}}$ and PSRO$_{\text{Nash}}$ start from scratch in each iteration, which poses a challenge for the algorithm to converge and introduces more variance in the training process.

Moreover, in PSRO algorithms, as the learned policy gradually improves with each iteration, the most recent policy of the population naturally poses greater difficulty for subsequent iterations. Therefore, PSRO$_{\text{Nash}}$ tends to put more weight on the most recent policy in the meta-strategy. This in turn has an effect on the learning of new policies. We can observe the outcomes in the heatmaps: for each row, the win rate against the most recent policy is often higher than the others. In FSP, on the other hand, the win rate against each policy is more evenly distributed, indicating that the population is potentially more balanced and stable.

## F.6    Low-level drills of hierarchical policy

Low-level drills are derived through PPO training, while the high-level skill is implemented as a rule-based, event-driven policy that determines which drone utilizes which skill in response to the current game state. In accordance with the *3 vs 3* task setting, each team consists of three drones positioned as front-left, front-right, and backward within their half of the court. Below, we describe each low-level drill and explain when it is utilized by the high-level policy.

***Hover.***    The *Hover* skill is designed to enable the drone to hover around a specified target position. This skill takes a three-dimensional target position as input. The skill is frequently utilized by the high-level policy. For instance, in the serve scenario, only the serving drone uses the *Serve* skill, while the other two teammates use the *Hover* skill to remain at their respective anchor points.

***Serve.***    The *Serve* skill is designed to enable the drone to serve the ball towards the opponent's side of the court. In accordance with the *3 vs 3* task setting, for the *Serve* skill, the ball is initialized at a position $3\,\mathrm{m}$ directly above the serving drone, with zero initial velocity. This skill is exclusively utilized by the high-level policy during the serve scenario, during which the designated serving drone employs the *Serve* skill at the start of a match.

***Pass.*** The *Pass* skill is designed to handle the opponent's serve or attack by allowing the drone to make the first contact of the team's turn and pass the ball to a teammate. This skill is exclusively used by the backward drone responsible for hitting the ball to the front-left teammate. The high-level policy designates the backward drone to utilize this skill whenever the opponent hits the ball.

***Set.*** The *Set* skill is designed to transfer the ball from the passing drone to the attacking drone, serving as the second contact in the team's turn. In our design, the front-left drone utilizes the *Set* skill to pass the ball to the front-right drone. The high-level policy designates the front-left drone to utilize this skill whenever the backward drone successfully makes contact with the ball.

***Attack.*** The *Attack* skill is designed to hit the ball towards the opponent's court, serving as the third and final contact in the team's turn. This skill includes a one-hot target input that specifies whether to direct the ball to the left side or the right side of the opponent's court. In our design, the front-right drone uses the *Attack* skill to strike the ball. The high-level policy assigns the front-right drone to utilize this skill whenever the front-left drone successfully hits the ball.

