# OpenReview forum: "VolleyBots: A Testbed for Multi-Drone Volleyball Game Combining Motion Control and Strategic Play"
_NeurIPS.cc/2025/Datasets_and_Benchmarks_Track — NeurIPS 2025 Datasets and Benchmarks Track poster_

### Official Review · Reviewer_1bMd · 2025-07-02

**Rating:** 5
**Confidence:** 3

**Summary:**

VolleyBots built a multi-drone volleyball simulation environment to research on reinforcement learning. This benchmark combines agile 3D flight control and multi-agent strategy and provides a suite of single/multi-agent cooperative and competitive tasks. This paper also provided some baseline policies. While the paper is interesting and offers a fresh direction for embodied multi-agent learning, I would encourage the authors to more clearly articulate the novelty of VolleyBots compared to existing benchmarks such as the RoboCup Soccer Simulation 3D League, and to elaborate on the potential benefits and challenges of bridging this simulation with real-world robotic experiments.

**Dataset Code Accessibility:**

Yes

**Ethical Considerations:**

No, there are no or only very minor ethics concerns

**Final Justification:**

The authors addressed my concerns about the impact of this simulation on the real experiment. Their zero-shot sim-to-real transfer also demonstrated the fidelity of their simulation environments. Therefore, I think this work deserves to be accepted.

**Limitations Weaknesses:**

* It is not clear how this simulated volleyball setting benefits the real-word robotics applications.
* The authors don't present a motivating real scenario beyond "sports as testbed".
* The real experiments for this setting may be difficult since the volleyball is heavy and drones are not designed for repeated impacts with hard objects. Therefore, it is hard to understand the gap between simulation and real experiment.
* RoboCup Soccer Simulation 3D is a simulation benchmark that also combines the robot dynamic control and the mixed cooperation and competition strategies of multiple-agents. RoboCup Soccer Simulation 3D looks more realistic and can be validated in RoboCup real experiments. It is unclear about the advantages of this benchmark compared to RoboCup Soccer simulation except changing the sport from football to volleyball.

**Strengths Contributions:**

* This is the first testbed for drone-based team sports with both cooperation and competition.
* This paper includes a clear curriculum tasks: from solo to full-team.
* This paper uses a physically accurate simulator to make the drone/ball dynamics realistic.
* Baseline and simulation code are provided, which is helpful to reproduce the experiments.
* Some benchmarks, such as Google Research Football, don't include the motion control of the agent. This paper combines both high-level decision-making and low-level motion control, which would allow the robot to learn both the high-level and low-level policies. Moreover, it may be easier to learn since the control of drones is easier than that of humanoid robot.
* This benchmark may help to improve the learning of multi-agent control strategies for drones, which may benefits the research of coordination under high-speed, low-latency control.

---

> ### Author Rebuttal · Authors · 2025-07-31
>
> We sincerely thank the reviewer for the constructive comments; below we respond to each comment point by point.
>
> > Q1: It is not clear how this simulated volleyball setting benefits the real-word robotics applications.
>
> Firstly, we have already demonstrated zero-shot Sim2Real transfer on the *Solo Bump* task. In hardware, a single drone achieved **154 consecutive bumps**, whereas—to our knowledge—the best previously reported result for single-drone bumping using traditional controllers is **14 bumps** [1][2]. A video of this real-world experiment is available on our project website, which is linked in the manuscript’s Abstract. We detail the full Sim2Real pipeline in our response to Q3. Taken together, these results indicate that the testbed is sufficiently high-fidelity and that policies trained in simulation have strong potential to transfer to real hardware.
>
> Secondly, in Sec. 2 Related work, we analyze in detail how our testbed differs from existing robotics-sport settings. Unlike existing tasks, ours features competitive and cooperative gameplay, a turn-based interaction structure, and agile 3D maneuvering. We will also provide in the response to Q4 a detailed comparison between our drone volleyball task and RoboCup Soccer. In addition, in Sec. 2.2 we explain in detail how our tasks differ from existing UAV tasks, posing distinct challenges for algorithms in the drone domain.
>
> [1] "Quadrocopter ball juggling." IROS, 2011.
>
> [2] "Ball juggling with an under-actuated flying robot." IROS, 2015.
>
> > Q2: The authors don't present a motivating real scenario beyond "sports as testbed".
>
> Competitive milestones have repeatedly driven AI progress—e.g., AlphaGo [1] in Go and OpenAI Five [2] in Dota 2. In embodied AI, recent examples include DeepMind humanoid football [3] and DeepMind robot arm table-tennis [4]. Along this trajectory, we introduce drone volleyball as a new task. It differs from prior settings and features *mixed competitive and cooperative game dynamics, turn-based interaction, agile 3D maneuvering*.
>
> Our primary goal is to provide a high-fidelity testbed that advances embodied AI research. In the long run, if onboard power is sufficient to play with a real volleyball (we currently use a small rubber ball in the real-world experiment), it could serve as an training partner for skill practice. More broadly, these skills—dynamic interception, controlled contact, and short-range redirection—align with UAV logistics handoffs, and we hope the testbed can inspire such follow-on research.
>
> [1] "Mastering the game of Go with deep neural networks and tree search." nature.
>
> [2] "Dota 2 with large scale deep reinforcement learning." arXiv, 2019.
>
> [3] "From motor control to team play in simulated humanoid football." Science Robotics.
>
> [4] "Achieving Human Level Competitive Robot Table Tennis." ICLR workshop, 2025.
>
> > Q3: The real experiments for this setting may be difficult since the volleyball is heavy and drones are not designed for repeated impacts with hard objects. Therefore, it is hard to understand the gap between simulation and real experiment.
>
> Firstly, regarding the physical feasibility. We would like to clarify that a standard volleyball was not used in our experiments. Acknowledging the drone's payload limitations, we selected a 47g, 4cm-diameter rubber ball, which is well within the capabilities of our 1100g quadrotor with its 5:1 thrust-to-weight ratio. To handle repeated impacts, the racket was also rigidly fastened to the drone's frame using custom aluminum columns to ensure hardware robustness.
>
> Furthermore, you correctly identify that impacts induce severe dynamic disturbances. Leveraging NVIDIA Isaac Sim, our simulation explicitly models the sharp, post-impact spikes in attitude and angular velocity. Therefore, instead of simplifying or avoiding this problem, our approach requires the RL agent to learn how to actively recover from these challenging disturbances, which is a core feature of our benchmark's difficulty.
>
> Ultimately, we have deployed the policy trained in our *solo bump* task with additional domain randomization and observation latency onto a **physical quadrotor** in a **zero-shot** transfer. The drone and the ball are tracked via a motion capture system, with velocity estimation using Kalman filters. The policy run onboard on an NVIDIA Orin NX platform. This successful zero-shot Sim2Real transfer not only demonstrates the **capability of RL**, but also provides strong evidence of **our testbed’s high fidelity and Sim2Real potential**.
>
> > Q4: RoboCup Soccer Simulation 3D is a simulation benchmark that also combines the robot dynamic control and the mixed cooperation and competition strategies of multiple-agents. RoboCup Soccer Simulation 3D looks more realistic and can be validated in RoboCup real experiments. It is unclear about the advantages of this benchmark compared to RoboCup Soccer simulation except changing the sport from football to volleyball.
>
> We acknowledge the strong impact and contributions of RoboCup Soccer. **Our aim, however, is not merely to change the sport from soccer to volleyball, but to target a different set of challenges while retaining Sim2Real potential.** As noted earlier (see our responces to Q1 & Q3), we have preliminary zero-shot Sim2Real results, indicating that our testbed offers comparable Sim2Real promise. Below we highlight the key differences between our drone-volleyball task and RoboCup Soccer:
>
> * Hardware platform. Our benchmark uses quadrotors, whereas RoboCup Soccer 3D uses humanoid robots. Quadrotors exhibit agile, fully 3D maneuvering and underactuated flight dynamics with rapid attitude–position coupling and impact-induced disturbances, which pose distinct algorithmic challenges relative to legged/humanoid control.
> * Interaction structure. Our task is turn-based with a strict limit on touches per side; in soccer, the ball can be contacted repeatedly to iteratively adjust its state. Consequently, in drone volleyball, agents must achieve high-precision, single-contact redirection, making accuracy at contact far more critical.
> * Simulation throughput. Leveraging NVIDIA Isaac Sim with a GPU-accelerated physics pipeline that largely removes CPU bottlenecks, our simulator can reach ~$10^5$ FPS on an RTX 4090—orders of magnitude higher than typical RoboCup Soccer Simulation 3D—which substantially accelerates policy training.
>
> These differences mean our testbed explores a different capability space and failure landscape, rather than serving as a sport-only variant of RoboCup Soccer.

---

> > ### Comment · Reviewer_1bMd · 2025-08-03
> >
> > Thank authors for addressing my concerns. I am happy to increase score from 4 to 5.

---

> > ### Author Response · Authors · 2025-08-03
> >
> > We appreciate the reviewer’s positive feedback and the increased score, and are glad our rebuttal addressed the concerns.

---

### Official Review · Reviewer_raGc · 2025-07-02

**Rating:** 5
**Confidence:** 4

**Summary:**

The paper presents a simulation platform named VolleyBots that allows the training and testing of drone control algorithms in multiple task scenarios. These scenarios include three single-agent tasks, three multi-agent cooperative tasks, and three multi-agent competitive tasks. In addition to that, the paper implemented a wide range of well-known RL algorithms and conducted an extensive validation of their performance on the proposed platform. The paper also proposes a simple hierarchical RL policy that outperforms the aforementioned methods.

**Dataset Code Accessibility:**

Yes

**Ethical Considerations:**

No, there are no or only very minor ethics concerns

**Final Justification:**

The author has answered and made explanations to my questions well. I can see no other significant problems in the paper, and it is suitable to be accepted.

**Limitations Weaknesses:**

1.It seems unfair to compare the end-to-end RL baselines with the hierarchical RL policy proposed by the paper (see section 5.4). This indeed gives an insight that a hierarchical control architect performs better at complex tasks, but it can hardly suffice as a convincing contribution, since the conclusion is somewhat obvious.

2.The description of the defense racket dynamic (see appendix C.3) is somewhat unclear. How is the position, attitude angles and velocity related to the drone carrying it? Will the attitude of the drone affect the attitude of the racket? The author should give a more detailed explanation on the matter to make it more understandable.

3.The current simulation environment seems deterministic and no disturbances exist. It is suggested that the authors can add an optional random disturbance/uncertainty to the system dynamic to further test the RL method’s robustness and adaptation ability against difference scales of disturbances/uncertainties.

**Strengths Contributions:**

1.The paper proposes and implements a wide range of task settings featuring multi or single drone volleyball. This is an interesting task for RL-based drones, which poses high challenge to the agents’ ability to perform rapid and accurate movements, as well as a swarm’s ability to cooperate or compete with each other. Simulation task scenarios of this kind is uncommon for drones at the current moment.

2.The paper performs a highly detailed and comprehensive comparison on the designed volleyball task scenarios between some of the existing RL algorithms and game-theoretic algorithms including QMIX, MADDPG, MAPPO, etc. (see appendix E). The result gives a clear assessment of their capabilities and drawbacks. It also provides the detailed parameters of the implemented algorithms, ensuring the reproductibility of the work.

3.The paper is well-organized and well-written. The work is solid and the content is profound.

---

> ### Author Rebuttal · Authors · 2025-07-31
>
> We appreciate the reviewer’s recognition of our work; below we provide point-by-point responses to your questions.
>
> > 1.It seems unfair to compare the end-to-end RL baselines with the hierarchical RL policy proposed by the paper (see section 5.4). This indeed gives an insight that a hierarchical control architect performs better at complex tasks, but it can hardly suffice as a convincing contribution, since the conclusion is somewhat obvious.
>
> Thank you for the thoughtful concern. Our intent is **not** to claim a generic superiority of hierarchical control over end-to-end learning, but to provide an **insight specific to our turn-based task** and to offer a task-aligned hierarchical design as **a stronger baseline for future work**.
>
> Drone volleyball is explicitly turn-based with a hard limit on touches per side; unlike soccer, agents cannot iteratively refine the ball state through repeated contacts. This demands single-contact precision at the moment of impact.
> Thus, in contrast to conventional “always-on” hierarchies, our high level is event-driven—it only triggers on discrete events (e.g., the moment drone hit the ball)—while the low-level controller acts every simulation step. This decouples decision frequency from control execution, reducing interference between long-horizon intent and fast attitude control, and is designed to improve contact accuracy under impact-induced disturbances.
>
> > 2.The description of the defense racket dynamic (see appendix C.3) is somewhat unclear. How is the position, attitude angles and velocity related to the drone carrying it? Will the attitude of the drone affect the attitude of the racket? The author should give a more detailed explanation on the matter to make it more understandable.
>
> Apologies for our bad wording, and we will explain it more explicitly in our updated manuscript. In fact, the racket in Appendix C.3 and the one mounted on the drone serve different functions.
>
> * **Defensive racket (Appendix C.3).** This racket is only utilized in the *set and spike (hard)* task. It is an environmental, rule-based element that returns the ball struck by the drone back to the drone’s side. It is not attached to the drone and has no kinematic coupling with it. A demo of Set-and-Spike (hard) is available on our project website linked in the manuscript’s Abstract.
>
> * **Racket mounted on the drone.** This racket is rigidly attached to the drone. Concretely, its pose is defined in the drone’s body frame with a fixed offset along the body normal (z-axis) and its face aligned with the body normal. Consequently, the drone shares both the velocity and attitude angles of the mounted racket.
>
> > 3.The current simulation environment seems deterministic and no disturbances exist. It is suggested that the authors can add an optional random disturbance/uncertainty to the system dynamic to further test the RL method’s robustness and adaptation ability against difference scales of disturbances/uncertainties.
>
> Thank you for the suggestion—we agree that disturbances are important, and we have incorporated optional stochasticity into the simulator. Specifically, we add domain randomization (over initial states and the ball–racket coefficient of restitution) and observation latency. We have deployed the policy trained in our *solo bump* task with these disturbances onto a **physical quadrotor** in a **zero-shot** transfer. Our platform consisted of a 1.1kg quadrotor (including the rigidly fastened racket) and a 47g, 4cm-diameter rubber ball. They are tracked via a motion capture system, with velocity estimation using Kalman filters. The policy run onboard on an NVIDIA Orin NX platform. The drone ultimately achieved **154 consecutive bumps**, while the best-reported result for single drone juggling using traditional control methods was only **14 bumps** [1][2]. A video of this real-world experiment is available on our project website, which is linked in the manuscript’s Abstract. This successful zero-shot Sim2Real transfer not only demonstrates the **capability of RL**, but also provides strong evidence of **our testbed’s high fidelity and Sim2Real potential**.
>
> [1] "Quadrocopter ball juggling." IROS, 2011.
>
> [2] "Ball juggling with an under-actuated flying robot." IROS, 2015.

---

> > ### Comment · Reviewer_raGc · 2025-08-04
> >
> > The author has answered and made explanations to my questions well. I can see no other significant problems in the paper, and it is suitable to be accepted.

---

> > ### Author Response · Authors · 2025-08-04
> >
> > Thank you for the careful evaluation and positive assessment. We are pleased that our clarifications addressed your concerns and appreciate the recommendation for acceptance.

---

### Official Review · Reviewer_sZm2 · 2025-07-06

**Rating:** 4
**Confidence:** 4

**Summary:**

This work introduces VolleyBots, a benchmark environment for multi-drone volleyball that requires motion control and strategic play. The benchmark encompasses single-agent, cooperative, competitive, and mixed scenarios, and the authors provide a series of curriculum tasks for future research. The benchmarking suite comprises tests for single-agent RL, multi-agent RL, and game-theoretical algorithms.

**Additional Feedback:**

Questions:

1. Beyond the inherent challenges of drone control, what is the practical utility or value in using drones for volleyball?
2. Given the provided drone dynamics model—which appears to treat drones as point masses or simplified balls—does this environmental simplification reduce the benchmark's difficulty compared to Gazebo-based robotics control tasks?
3. Despite thorough training, PPO significantly outperforms SAC and TD3 in both performance and learning speed (Fig. 6), even in single-agent tasks. This contradicts established robotics benchmarks. Could the authors analyze this discrepancy? Is it attributable to SAC/TD3's sensitivity to hyper-parameter tuning?
4. In Table 4, do the symbols $\uarr$ and $\darr$ denote "higher-is-better"/"lower-is-better" or represent performance deltas?
5. Were the hyperparameters used in experiments obtained via systematic search (e.g., grid search) or transferred from other domains? Could task-specific adjustments to hyper-parameters and policy network architectures improve results?
6. Appendix F.1 reports low computational overhead (using Isaac Sim). Could this enable more extensive hyper-parameter searches and additional random seeds (beyond the current three) to strengthen statistical confidence?
7. Is the distinction between FSP and PSRO$_\text{uniform}$ solely that FSP inherits policies from prior iterations? Per FSP theory, convergence is defined by the average strategy—not the latest policy. Which policy was evaluated for FSP in this work?



Typos:

Line 194: The second Spike (Easy) should be Spike (Hard)

**Dataset Code Accessibility:**

Yes

**Dataset Code Comments:**

The authors provide sufficient details in the main text and offer corresponding code. I have verified the provided URL, which includes environment setup instructions and pre-trained PyTorch models for various tasks. (Note: I did not explicitly confirm whether the models can be successfully loaded or if their performance aligns with the results reported in the paper.)

**Ethical Considerations:**

No, there are no or only very minor ethics concerns

**Final Justification:**

The detailed training settings make the experimental results more solid. Besides, the discussions will also improve the paper.

**Limitations Weaknesses:**

The statistical rigor of the benchmark conclusions is somewhat limited. As indicated in Appendix F, experiments were computationally efficient: single-agent runs completed in 12 hours, and multi-agent runs in 24 hours on a workstation. This suggests relatively low computational demands (Additionally, Isaac Sim is known for high speed and massive parallelism). However, the conclusion confidence is undermined by the use of only three random seeds and an unclear hyper-parameter search space. Furthermore, the observation that PPO significantly outperformed SAC and TD3 in single-agent settings diverges from established patterns in MuJoCo and similar continuous control benchmarks. A similar anomaly appears in multi-agent results, where MAPPO substantially outperformed HAPPO.



From my perspective, another limitation concerns the task's practical relevance. The real-world applicability of multi-drone volleyball (VolleyBots) is less apparent. While the authors justify it by combining the inherent challenges of drone control and the need for collaboration in volleyball, the connection to tangible robotics applications remains questionable. It would be more compelling if the benchmark tasks were grounded in practical multi-drone problems, such as logistics (e.g., coordinated delivery) or infrastructure inspection.

**Strengths Contributions:**

This paper is well-organized and easy to follow. The proposed VolleyBots environment presents a dual challenge, encompassing both low-level motion control and high-level strategic play. Drawing inspiration from typical scenarios in human volleyball, the authors designed analogous situations for drones. These include fundamental striking and coordinated maneuvers, as well as scenarios requiring complex teamwork. Furthermore, the authors developed a diverse set of tasks within VolleyBots targeting both single-agent and multi-agent systems. This suite ranges from simple to complex control tasks, features inter-drone cooperation and inter-team competition, establishing a novel platform for future robotics research.



Additionally, the authors provide a relatively detailed description of the training setup and process. The evaluation of algorithms like PSRO is technically sound—a crucial aspect often overlooked in many existing environments. The figures presenting the results are also clear and effective. Experimental results demonstrate that standard RL algorithms can learn policies achieving competent performance. However, performance degrades significantly as the number of agents increases.



Overall, VolleyBots represents a notably challenging and engaging benchmark for robotics learning community.

---

> ### Author Rebuttal · Authors · 2025-07-31
>
> We thank the reviewer for the constructive feedback. We respond point-by-point to your comments below. For clarity, we label Limitation Weaknesses with the prefix “L” and questions with “Q”. We hope these clarifications address your concerns.
> > L1 & Q3 & Q5 & Q6: About baseline performance
>
> **1. How did we derive the baseline hyperparams used in out manuscript?**
>
> 1.1 Search space
>
> Shared hyperparams:
> * Batchsize: [2k, 4k, 8k, 16k]
> * Actor lr: [5e-3, 5e-4, 5e-5]
> * Critic lr: [5e-3, 5e-4, 5e-5]
> * Max grad norm: [5, 10, 20]
> * MLP architecture: [(256, 128), (256, 128, 128)]
> * Gamma: [0.95, 0.99, 0.995]
>
> Shared off-policy hyperparams:
> * Buffer size: [5e5, 1e6]
> * Gradient steps: [0.25k, 0.5k, 1k, 2k]
> * Tau (soft update): [1e-3, 5e-3, 1e-2]
>
> Hyperparams for DDPG:
> * Policy noise: [0.1, 0.2]
> * Noise clip: [0.2, 0.4]
>
> Hyperparams for TD3:
> * Target noise: [0.1, 0.2]
> * Policy noise: [0.1, 0.2]
> * Noise clip: [0.2, 0.4]
> * Actor delay: [2, 4]
>
> Hyperparams for SAC:
> * Alpha learning rate: [1e-4, 5e-4]
>
> Hyperparams for PPO:
> * PPO epochs: [1, 2, 4, 8]
> * PPO clip ratio: [0.1, 0.2]
> * GAE lambda: [0.9, 0.95, 0.99]
>
> 1.2 Search protocol and result
>
> Because the joint search space is high-dimensional and an exhaustive grid search is prohibitive, we used **random search** over the space above. We tuned on a *Hover* task, where the drone starts from randomized position/heading, moves to a randomized target, and then maintains stable position and heading without drift. The reward is related to position error, heading alignment, uprightness, and spin. The best configurations found on *Hover* yield the following baseline performance (over 3 seeds):
>
> |Algorithms|Return|
> |-|-|
> |DDPG|1168.01 ± 16.83|
> |TD3|1212.82 ± 8.81|
> |SAC|1249.39 ± 4.55|
> |PPO|1196.68 ± 3.01|
>
> On this simple task, DDPG, TD3, SAC, and PPO perform comparably. **We fixed the per-algorithm configuration selected on *Hover* and transferred it unchanged to all other tasks in the testbed as reported in the original manuscript. This was to ensure cross-task fairness and reproducibility, and to evaluate robustness without task-specific tuning.** As reported in Table 2 of the manuscript, PPO outperforms DDPG, TD3, and SAC on our proposed tasks, suggesting that, in this setting, **PPO is more robust to hyperparams** than the off-policy methods.
>
> **2. Task-specific grid search**
>
> To address Q5, given the limited rebuttal window, we conducted **task-specific grid searches** for TD3 with PRT action space. Based on pilot experiments, we chose three core hyperparams:
> * Gradient steps: [0.5k, 1k, 2k]
> * (policy noise, noise clip): [(0.1, 0.2), (0.2, 0.4)]
> * Actor delay: [1, 2, 4]
>
> We highlight the best per-task hyperparameter in bold in the following tables.
>
> 2.1 *Back and Forth*
>
> |gradient_steps|**(policy noise, noise_clip)**|**actor_delay**|result|
> |-|-|-|-|
> |0.5k|(0.1, 0.2)|1|0.98|
> |||2|0.98|
> |||4|0.96|
> ||(0.2, 0.4)|1|0.99|
> |||2|0.96|
> |||4|1.00|
> |**1k**|(0.1, 0.2)|1|0.94|
> |||2|0.99|
> |||4|0.97|
> ||**(0.2, 0.4)**|1|0.94|
> |||2|0.96|
> |||**4**|**6.58**|
> |2k|(0.1, 0.2)|1|0.98|
> |||2|0.92|
> |||4|2.12|
> ||(0.2, 0.4)|1|0.98|
> |||2|0.97|
> |||4|4.59|
>
> 2.2 *Hit the Ball*
>
> |gradient_steps|**(policy noise, noise_clip)**|**actor_delay**|result|
> |-|-|-|-|
> |0.5k|(0.1, 0.2)|1|3.00|
> |||2|4.14|
> |||4|3.58|
> ||(0.2, 0.4)|1|3.75|
> |||2|0.40|
> |||4|3.92|
> |**1k**|**(0.1, 0.2)**|1|3.48|
> |||2|3.81|
> |||**4**|**4.93**|
> ||(0.2, 0.4)|1|3.90|
> |||2|4.09|
> |||4|3.86|
> |2k|(0.1, 0.2)|1|3.94|
> |||2|3.98|
> |||4|3.87|
> ||(0.2, 0.4)|1|3.83|
> |||2|3.87|
> |||4|3.79|
>
> 2.3 *Solo Bump*
>
> |gradient_steps|**(policy noise, noise_clip)**|**actor_delay**|result|
> |-|-|-|-|
> |0.5k|(0.1, 0.2)|1|4.06|
> |||2|5.42|
> |||4|2.35|
> ||(0.2, 0.4)|1|4.57|
> |||2|2.82|
> |||4|5.84|
> |1k|(0.1, 0.2)|1|6.85|
> |||2|4.45|
> |||4|5.62|
> ||(0.2, 0.4)|1|4.97|
> |||2|5.89|
> |||4|5.25|
> |**2k**|**(0.1, 0.2)**|1|7.10|
> |||**2**|**8.80**|
> |||4|6.56|
> ||(0.2, 0.4)|1|7.01|
> |||2|2.06|
> |||4|4.15|
>
> 2.4 Analysis
>
> Our grid search tuned only three hyperparams, but performance varied widely within each task, and with well-chosen hyperparams TD3 **substantially outperformed** our earlier TD3 performance in the manuscript.
> Note that we tuned only three hyperparams for TD3, and under this minimal search it still did not reach the PPO performance without per-task tuning in the manuscript.
> We also observed that **the best-performing hyperparams differed from task to task**. These observations indicate that task-specific grid search is helpful, yet they also reinforce that, compared with PPO, off-policy methods are **less robust** to hyperparameter choices.
>
> **3. Why are PPO’s hyperparams more robust for our tasks?**
>
> Different from Mujoco, our volleyball environment is **turn-based** and **event-driven**, which creates long-term dependencies and rewards that are dominated by **discrete events**. e.g. in *Solo Bump* most reward is obtained only when a bump succeeds; between events, rewards are near zero at each step. This makes the effective learning signal spiky and highly sensitive to exploration choices. Off-policy methods must learn a critic from replayed, off-policy data; in this setting, bootstrapping error interacts with exploration noise, which increases hyperparameter sensitivity. By contrast, PPO’s on-policy updates, clipped surrogate objective, and GAE reduce update volatility and distribution shift, yielding a wider, more forgiving hyperparameter region on our testbed.
>
> Engineering tricks, such as prioritized replay buffer, annealed exploration noise, and n-step returns may improve off-policy methods performance. However, we intentionally keep baselines simple and comparable in the main table.
>
> **4. Experimental differences between MAPPO and HAPPO.**
>
> In Table 3 of the manuscript, across six tasks, HAPPO outperforms MAPPO on one task, is comparable on two, is slightly worse on two, and is significantly worse on one. Overall, this indicates that MAPPO holds a slight overall advantage over HAPPO on our testbed, rather than “substantially outperforming” it.
>
> **5. Why did we choose three seeds?**
>
> In Fig. 6–7, the inter-seed variance is small on our testbed. Pilot experiments indicated diminishing gains beyond three seeds, so we chose three to balance statistical power with coverage across tasks and algorithms under a fixed compute budget.
> > L2 & Q1 & Q2: About real-world applications
>
> **1. Sim2Real promise**
>
> The drone model is not a simplified model, but a standard 6-DoF rigid-body dynamics model, with an inertia tensor, coupled orientation, and rotational dynamics (Appendix C.2). Also, we simulated the motor dynamics and replicated the low-level controller consistent with the PX4 flight control.
>
> We have deployed the policy trained in our *solo bump* task with additional domain randomization and observation latency onto a **physical quadrotor** in a **zero-shot** transfer. Our platform consisted of a 1.1kg quadrotor (including the rigidly fastened racket) and a 47g, 4cm-diameter rubber ball. They are tracked via a motion capture system, with velocity estimation using Kalman filters. The policy run onboard on an NVIDIA Orin NX platform. The drone ultimately achieved **154 consecutive bumps**, while the best-reported result for single drone juggling using traditional control methods was only **14 bumps** [1][2]. A video of this real-world experiment is available on our project website, linked in the manuscript’s Abstract. This Sim2Real transfer not only demonstrates the **capability of RL**, but also provides strong evidence of **our testbed’s high fidelity and Sim2Real potential**.
>
> [1] "Quadrocopter ball juggling." IROS, 2011.
>
> [2] "Ball juggling with an under-actuated flying robot." IROS, 2015.
>
> **2. Practical value**
>
> Competitive milestones have repeatedly driven AI progress—e.g. AlphaGo in Go and OpenAI Five in Dota 2. In embodied AI, recent examples include DeepMind humanoid football [1] and DeepMind robot arm table-tennis [2]. Along this trajectory, we introduce drone volleyball as a new task. It differs from prior settings and features *mixed competitive and cooperative game dynamics, turn-based interaction, agile 3D maneuvering*.
>
> Our primary goal is to provide a high-fidelity testbed that advances embodied AI research. In the long run, if onboard power is sufficient to play with a real volleyball (we currently use a small rubber ball in the real-world experiment), it could serve as an training partner for skill practice. More broadly, these skills—dynamic interception, controlled contact, and short-range redirection—align with UAV logistics handoffs, and we hope the testbed can inspire such follow-on research.
>
> [1] "From motor control to team play in simulated humanoid football." Science Robotics.
>
> [2] "Achieving Human Level Competitive Robot Table Tennis." ICLR workshop, 2025.
> > Q4: Detail of table 4
>
> In Table 4, the symbols indicate directionality—“higher is better” or “lower is better”—and do not represent performance deltas. We will make this explicit in the revised manuscript.
> > Q7: Detail of FSP and $PSRO_{uniform}$.
>
> **1. On the distinction between FSP and $PSRO_{uniform}$.**
>
> In our implementation, the only difference between FSP and PSRO is initialization. The original Fictitious Self-Play (FSP) [1] was formulated for extensive-form games. A recent FSP formulation [2] lifts the extensive-form game to a meta-game which focuses on the selection of policies rather than isolated actions. We adopt this view, so our FSP implementation is operationally identical to PSRO that iteratively trains new oracles from scratch that approximate best responses; the only difference is the oracle initialization scheme.
>
> **2. Which policy is evaluated for FSP.**
>
> Consistent with FSP theory, we evaluate the average strategy induced by the learned population—by uniformly sampling a policy from the population for each game—rather than the single latest policy.
>
> [1] "Fictitious self-play in extensive-form games." ICML, 2015.
>
> [2] "A survey on self-play methods in reinforcement learning." arXiv, 2024.

---

> > ### Author Response · Authors · 2025-08-08
> >
> > Dear reviewer,
> >
> > We have proactively extended our experiments from three seeds to five seeds to further strengthen the robustness of our conclusions.
> > The two additional seeds for the single-agent tasks yield results highly consistent with our earlier findings.
> >
> >
> > Additional seeds results:
> > | algo | seed | Back and Forth-CTBR | Back and Forth-PRT | Hit-CTBR | Hit-PRT | Bump-CTBR | Bump-PRT |
> > |-|-|-|-|-|-|-|-|
> > | dqn | 4 |0.00 | 0.00 | 0.39 | 1.41 | 0.00 | 0.00 |
> > | dqn | 5 |0.00 | 0.00 | 0.38 | 1.62 | 0.00 | 0.00 |
> > | ddpg | 4 |0.98 | 0.86 | 2.47 | 4.06 | 0.14 | 0.14 |
> > | ddpg | 5 |1.75 | 1.10 | 3.56 | 4.42 | 0.64 | 0.80 |
> > | td3 | 4 |0.99 | 0.98 | 2.66 | 3.86 | 1.86 | 3.90 |
> > | td3 | 5 |1.62 | 0.99 | 3.06 | 4.13 | 5.29 | 5.94 |
> > | sac | 4 |0.83 | 0.77 | 4.18 | 6.31 | 0.88 | 2.18 |
> > | sac | 5 |0.72 | 0.42 | 2.83 | 5.96 | 0.68 | 1.02 |
> > | ppo | 4 |9.65 | 10.12 | 10.45 | 11.40 | 7.34 | 9.04 |
> > | ppo | 5 |9.33 | 10.35 | 10.57 | 11.33 | 9.51 | 11.67 |
> >
> >
> > Five-seed aggregated results:
> >
> > | algo | Back and Forth-CTBR | Back and Forth-PRT | Hit-CTBR | Hit-PRT | Bump-CTBR | Bump-PRT |
> > |-|-|-|-|-|-|-|
> > | DQN | 0.00 ± 0.00 | 0.00 ± 0.00 | 0.39 ± 0.02 | 1.88 ± 0.34 | 0.00 ± 0.00 | 0.00 ± 0.00 |
> > | DDPG | 1.14 ± 0.34 | 0.83 ± 0.23 | 2.87 ± 0.55 | 3.98 ± 1.08 | 0.44 ± 0.34 | 0.67 ± 0.32 |
> > | TD3 | 1.12 ± 0.68 | 0.99 ± 0.01 | 3.00 ± 0.52 | 3.91 ± 0.35 | 3.68 ± 1.43 | 5.29 ± 1.28 |
> > | SAC | 0.90 ± 0.12 | 0.83 ± 0.25 | 3.76 ± 1.46 | 3.87 ± 2.34 | 0.54 ± 0.27 | 1.36 ± 0.60 |
> > | PPO | 9.25 ± 0.31 | 10.04 ± 0.20 | 10.48 ± 0.08 | 11.40 ± 0.06 | 8.58 ± 0.79 | 10.83 ± 1.24 |
> >
> > Notably, the low standard deviations and consistent performance trends across all five seeds demonstrate the reliability of our findings.
> >
> > For completeness, we are also running the multi-agent experiments with five seeds.
> > Although they may not complete before the discussion period ends, we commit to including the full updated results in the next version of our manuscript.
> >
> > We believe these additional, more robust results directly address your concerns and further substantiate our claims.
> > We would greatly appreciate it if this strengthened evidence could be taken into consideration in your final evaluation.

---

### Comment · Area_Chair_nseJ · 2025-08-01
**All reviewers, please reply promptly**

Greetings paper 408 team,

We are now in the Author-Reviewer Discussions (July 31 - Aug 6), and for this paper, authors have provided detailed rebuttals to all reviews. Reviewer responses and discussion are a mandatory part of the process and will factor in to decisions about the Responsible Reviewing initiative, up to and including desk rejections for poor reviews or complete inactivity.

Beyond this, we seek to have an informative and helpful discussion as a key part of our community. Please write the author's back right away, doing more than just acknowledging you've read the rebuttal. Your goal should be to engage with the authors on points where you can help them improve the work or to add explanation when they don't well enough understand your scoring justification. Do reconsider your scores when new information is added, but of course it is OK to keep the same, but please with as much explanation as you can manage!

Thank you for your great help with NeurIPS this year,

Paper AC.

---

### Note · Authors · 2025-08-13

Dear reviewers, AC, SAC, and PC,

We would like to conclude our original paper and the rebuttal stage, and append additional experiment results addressing reviewer sZm2.

Our work introduces VolleyBots, a novel robotic sports testbed for the multi-drone volleyball game, combining precise motion control with high-level strategic play. It features competitive and cooperative gameplay, a turn-based interaction structure, and agile 3D maneuvering.
Inspired by how humans progressively learn volleyball, we design a curriculum of tasks ranging from single-drone drills to competitive matchups.
We implement representative RL, MARL, and game-theoretic baselines.
We also design a simple hierarchical policy that demonstrates a promising direction.

During the rebuttal phase, we add domain randomization and observation latency to our tasks. We also demonstrate the sim2real potential: in the *solo bump* task, our approach achieves 154 consecutive bumps via zero-shot sim2real transfer, substantially surpassing existing baselines. We clarify our hyperparameter selection methodology and report more multi-seed experiments.

Below are our multi-agent task results addressing reviewer sZm2:

|algo|seed|Bump and Pass w.o. shaping|Bump and Pass w. shaping|Set and Spike (Easy) w.o. shaping|Set and Spike (Easy) w. shaping|Set and Spike (Hard) w.o. shaping|Set and Spike (Hard) w. shaping|
|-|-|-|-|-|-|-|-|
|QMIX|4|0.09|0.09|0.02|0.02|0.02|0.02|
|QMIX|5|0.09|0.09|0.02|0.02|0.02|0.02|
|MADDPG|4|0.70|0.79|0.20|0.23|0.20|0.22|
|MADDPG|5|0.58|0.72|0.19|0.24|0.20|0.19|
|MAPPO|4|11.51|12.94|0.25|0.98|0.25|0.75|
|MAPPO|5|11.56|13.45|0.25|0.99|0.25|0.73|
|HAPPO|4|8.82|11.08|0.25|0.98|0.25|0.75|
|HAPPO|5|11.02|12.56|0.25|0.98|0.25|0.75|
|MAT|4|0.99|12.55|0.25|0.74|0.25|0.75|
|MAT|5|12.91|12.76|0.25|0.75|0.25|0.99|

Five-seed aggregated results:

|algo|Bump and Pass w.o. shaping|Bump and Pass w. shaping|Set and Spike (Easy) w.o. shaping|Set and Spike (Easy) w. shaping|Set and Spike (Hard) w.o. shaping|Set and Spike (Hard) w. shaping|
|-|-|-|-|-|-|-|
|QMIX|0.09 ± 0.01|0.09 ± 0.00|0.02 ± 0.00|0.02 ± 0.00|0.02 ± 0.00|0.02 ± 0.00|
|MADDPG|0.79 ± 0.15|0.84 ± 0.09|0.22 ± 0.02|0.23 ± 0.01|0.22 ± 0.02|0.22 ± 0.02|
|MAPPO|**11.32 ± 0.91**|**13.71 ± 0.58**|**0.25 ± 0.00**|**0.99 ± 0.00**|**0.25 ± 0.00**|0.75 ± 0.01|
|HAPPO|7.95 ± 3.67|12.14 ± 0.83|**0.25 ± 0.00**|0.98 ± 0.00|**0.25 ± 0.00**|0.79 ± 0.10|
|MAT|7.39 ± 6.0|13.11 ± 0.43|**0.25 ± 0.00**|0.89 ± 0.13|**0.25 ± 0.00**|**0.80 ± 0.11**|

---

### Decision · Program_Chairs · 2025-09-18

**Decision:**

Accept (poster)

**Comment:**

This paper presents a novel benchmark where the subject matter involves drone volleyball. This is an eye-catching demonstration that would surely draw a crowd at any public demonstration or media event, a positive. Some of the reviewer discussion has revolved around the centrality of the problem for a general machine learning community and the execution details.

Decision Justification: This paper seems to be technically strong and has an intriguing novelty. However, it does follow a very familiar pattern of robot sports – the intellectual surprise factor is low. I support a weak acceptance as a poster.

Detailed Meta-Review:

We are in a moment where many “toy” robotics tasks are known. Robo-cup and soccer variants have been well established for many decades. It is likely not the case that every combination of {robot modality}x{sport} needs to be available for experimentation to support the community. Will this one version of drone volleyball achieve wide use and add something that is not already possible with existing tools?

The authors highlight the novelty of having drones as the agents within each team as well as the nature of volleyball’s ruleset to distinguish from existing work, in addition to being more modern, based on parallelizable IsaacSim and transferable to real drones to impressive effect. In the end, the reviewers have largely been convinced of this argument. I also find it compelling but have a David-vs-Goliath concern – while VolleyBots may have some technical interest over RoboCup, it falls within a large shadow and is similar enough that, without a yearly community effort or some form of affiliation, the real impact is likely to be low.

The paper was judged easy to read by all reviewers. There were no concerns about data or code access or quality. Technical elements and experiments have largely been judged strong after discussion and refinement.

There has been some discussion about the ordering of baseline algorithm performance. Perhaps the PPO-TD3 ordering is unexpected by some reviewers. I recommend that we overlook this, as these are all modern algorithms with unique performance characteristics that depend on reward and dynamics structure. Although it is reasonable to ask for hyper-param optimization to be done well and transparently, we don’t want to force folks to wash out their real observed results at a particular stage of refinement, as this reality will also be faced by RL practitioners when setting up new environments. The authors have made a good argument in their rebuttal.

Reviewer sZm2 questioned the use of only 3 seeds in initial experiments, but 2 have additionally been added to strengthen statistics during discussions.

Reviewer raGc asked a very helpful question about stochasticity in the environment and the authors have added this during discussions. The authors discuss a sim2real result as the follow-on for this addition. While this is helpful, there are also questions about how stochastic execution would impact various learners and coordination hierarchy elements. I would encourage a further results table in the final supplemental to compare as many reported elements between deterministic/stochastic modes as is possible.

Reviewer 1bMd asked for specific comparison to Robocup. The authors’ most convincing distinction is the “three hits per side” nature of volleyball which rules out ongoing refinement by one team prior to action. This is noteworthy, but maybe not fundamental. A three-touch-before-shot version of soccer is likely also possible to setup, but the point is well taken that a variety of real world conditions are always helpful in ensuring ML methods generalize.